# Gradient-Based Multi-Objective Deep Learning: Algorithms, Theories, Applications, and Beyond

## Abstract

Many modern deep learning applications require balancing multiple objectives that are often conflicting. Examples include multi-task learning, fairness-aware learning, and the alignment of Large Language Models (LLMs). This leads to multi-objective deep learning, which tries to find optimal trade-offs or Pareto-optimal solutions by adapting mathematical principles from the field of Multi-Objective Optimization (MOO). However, directly applying gradient-based MOO techniques to deep neural networks presents unique challenges, including high computational costs, optimization instability, and the difficulty of effectively incorporating user preferences. This paper provides a comprehensive survey of gradient-based techniques for multi-objective deep learning, with a primary focus on supervised learning settings. We systematically categorize existing algorithms based on their outputs: (i) methods that find a single, well-balanced solution, (ii) methods that generate a finite set of diverse Pareto-optimal solutions, and (iii) methods that learn a continuous Pareto set of solutions. In addition to this taxonomy, the survey covers theoretical analyses, key applications, practical resources, and highlights open challenges and promising directions for future research.

## 1 Introduction

Traditional deep learning often focuses on optimizing a single learning objective, such as minimizing the prediction error or maximizing the likelihood. However, many real-world applications require balancing multiple, often conflicting, objectives. For instance, a computer vision system might need to perform the tasks of segmentation, depth estimation, and surface normal prediction simultaneously (Vandenhende et al., 2021), thus moving from single-task learning to multi-task learning (Zhang & Yang, 2022). Similarly, Large Language Models (LLMs) are expected to excel at diverse tasks like reasoning and coding, while also being safe, fair, and harmless (Wang et al., 2023a). Multi-Objective Optimization (MOO) (Miettinen, 1999), a field originating from operations research (Ehrgott, 2005), provides a formal framework for navigating these trade-offs. It has been widely studied across science and engineering, with diverse applications such as in finance (Steuer & Na, 2003), engineering design (Marler & Arora, 2004), and transportation planning (Tzeng et al., 2005).

Recently, there has been a surge of interest in adapting MOO for deep learning, re-framing many popular learning problems as multi-objective optimization problems. Examples include multi-task learning where each task performance is considered as an objective (Sener & Koltun, 2018), fairness-aware learning where accuracy is one objective and fairness metrics constitute the other objectives (Martinez et al., 2020), LLM alignment where each alignment criterion (such as helpfulness, harmlessness, and honesty) represents a separate objective (Wang et al., 2023a), and federated learning where the performance on each client is treated as an individual objective (Hu et al., 2022). All these problems can be formally expressed as:

$$\min_{\boldsymbol{\theta} \in \mathcal{K} \subset \mathbb{R}^d} \quad \boldsymbol{f}(\boldsymbol{\theta}) := [f_1(\boldsymbol{\theta}), \ldots, f_m(\boldsymbol{\theta})]^\top, \tag{1}$$

where $m \geq 2$ is the number of objectives, $\boldsymbol{\theta}$ is the decision variable, $\mathcal{K} \subset \mathbb{R}^d$ is the feasible set of the decision variable, and each $f_i : \mathcal{K} \to \mathbb{R}$ represents an individual objective function to be minimized. Here, minimizing the vector-valued function $\boldsymbol{f}$ should be understood in the standard vector-optimization sense (Boyd

& Vandenberghe, 2004; Miettinen, 1999): solutions are compared through Pareto dominance rather than a total scalar order, and concrete optimization procedures typically seek Pareto-optimal solutions or optimize scalarized subproblems as introduced in Section 2.

Unlike single-objective optimization, which focuses on finding a single best solution, MOO recognizes that no single solution is optimal for all objectives simultaneously. Instead, MOO aims to identify a set of solutions that represent different trade-offs between objectives, collectively known as the Pareto set (Miettinen, 1999). The Pareto set consists of all solutions where improving any objective would necessarily worsen at least one other objective, representing the best possible trade-offs available. In real-world applications, users often have varying preferences for these trade-offs. For instance, in the development of LLMs, a customer service application might emphasize harmlessness and safety, whereas an educational tool might prioritize reasoning and factual accuracy. To address this variability, users can specify their preferences using a vector, $\boldsymbol{\alpha} = [\alpha_1, \ldots, \alpha_m]^\top \in \Delta_{m-1}$, where $\Delta_{m-1} = \{\boldsymbol{\alpha} \in \mathbb{R}_+^m : \sum_{i=1}^m \alpha_i = 1\}$ is the probability simplex and each $\alpha_i$ represents the importance assigned to the $i$-th objective. MOO approaches can accommodate these user-defined preferences, enabling the discovery of solutions that align with individual needs.

MOO methods can be broadly divided into gradient-free and gradient-based approaches. Gradient-free methods, such as evolutionary algorithms (Deb et al., 2002; Zhang & Li, 2007) and particle swarm optimization (Coello et al., 2004), explore the solution space using population-based sampling. While effective for traditional low-dimensional MOO problems (Deb, 2011), they typically scale poorly when directly searching the parameter space of deep neural networks, which often contains millions or billions of parameters. This scope is different from recent evolutionary prompt-optimization methods for LLMs, such as EvoPrompt (Guo et al., 2024a) and GEPA (Agrawal et al., 2025), which operate in the discrete prompt space rather than optimizing all neural-network weights. This makes gradient-based methods (Désidéri, 2012; Mukai, 1980; Fliege & Svaiter, 2000; Liu et al., 2021a) preferable for deep neural networks, as they efficiently guide the search using objective gradients.

However, gradient-based MOO faces several significant challenges in deep learning. First, incorporating user preferences is difficult because simple scalarization can yield solutions that are poorly aligned with the specified preference vector, especially on non-convex Pareto fronts (Lin et al., 2019a; Mahapatra & Rajan, 2020; Navon et al., 2021). Second, training even one neural network is computationally expensive, posing a major challenge to efficiently approximating the entire Pareto set of networks (Navon et al., 2021; Chen & Kwok, 2024a; Zhong et al., 2024). Finally, mini-batch optimization introduces stochastic-gradient noise and biased common descent directions, which can destabilize gradient-based MOO and has motivated specialized stochastic analyses (Liu & Vicente, 2021; Zhou et al., 2022; Fernando et al., 2023; Chen et al., 2023c). These challenges have motivated extensive research efforts to develop specialized MOO techniques for deep learning. This paper provides a comprehensive survey of these advances, systematically categorizing existing approaches and identifying key open problems in the field.

## 1.1 Scope of the Survey

This survey focuses on gradient-based multi-objective optimization methods within the context of supervised deep learning. We concentrate on gradient-based approaches because, as discussed above, the high dimensionality of modern deep neural networks renders gradient-free methods impractical in these settings. Within this scope, we cover methods spanning the full spectrum of solution outputs: from finding a single balanced model, finding a finite set of Pareto-optimal solutions, to learning a continuous representation of the entire Pareto set. While multi-objective reinforcement learning is a closely related and highly active field, it possesses distinct characteristics and is already well-documented in existing comprehensive surveys (Felten et al., 2024; Roijers et al., 2013; Hayes et al., 2022). Consequently, although certain methods discussed herein can be adapted for MORL, algorithms designed exclusively for reinforcement learning fall outside the scope of this paper.

## 1.2 Survey Methodology

We collected papers from major machine learning venues, arXiv, OpenReview, Google Scholar, and references of relevant papers using keywords including "multi-objective optimization", "multi-objective deep

learning", "multi-task learning", "Pareto set learning", "Pareto optimization", and "gradient balancing". The cutoff date is March 31, 2026. We include works that formulate supervised deep learning problems as MOO problems and use gradient-based training for either a single solution, a finite solution set, or a continuous Pareto-set approximation. We exclude multi-objective optimization methods designed exclusively for reinforcement learning, Bayesian optimization, or purely gradient-free/evolutionary search.

### 1.3 Comparison with Related Works

Previous literature has explored various facets of multi-objective optimization and multi-task learning, though none fully address the specific intersection of gradient-based methods and deep learning. Foundational books (Miettinen, 1999; Ehrgott, 2005) and general surveys (Eichfelder, 2021) provide introductions to classic MOO from a broad optimization perspective, but they primarily focus on early methods and lack an emphasis on deep learning. Other reviews, such as Wei et al. (2021), focus extensively on evolutionary MOO algorithms. While effective in lower-dimensional spaces, these gradient-free approaches are generally unsuitable for deep neural networks due to the massive parameter spaces.

In the machine learning domain, several surveys explore multi-task learning. Zhang & Yang (2022) provide an overview of traditional multi-task methods, while Crawshaw (2020) summarizes deep multi-task learning advancements up to 2020. More recently, Yu et al. (2024) presented a comprehensive review spanning both traditional and deep multi-task learning. These surveys usually take tasks, sharing architectures, and learning paradigms as the organizing units, and mainly ask how to train one shared model or task-head system that performs well across tasks. The MOO perspective changes the problem object: it treats the loss vector and its partial order as central, so there may be no single scalar optimum before preferences are specified. Consequently, MOO methods are organized by whether they return one compromise, a finite Pareto-set approximation, or a preference-conditioned continuum, and they are evaluated through Pareto dominance/stationarity, hypervolume, preference alignment, and coverage of trade-offs. This is why works such as preference-vector finite-set methods (Lin et al., 2019a; Mahapatra & Rajan, 2020; Zhang et al., 2024b), hypervolume or particle-based Pareto-set methods (Wang et al., 2017; Liu et al., 2021d), and continuous Pareto-set learning methods (Navon et al., 2021; Chen & Kwok, 2024a; Zhong et al., 2024) are not naturally covered by standard MTL surveys.

The most closely related work to ours is by Peitz & Hotegni (2024), who survey MOO algorithms for deep learning. However, their work focuses on general MOO algorithms and covers only a limited selection of gradient-based methods. Crucially, it lacks a detailed structural taxonomy of gradient-based approaches and omits in-depth discussions on theoretical foundations and practical applications. Finally, several other surveys concentrate strictly on the application of MOO in specific domains, such as dense prediction tasks (Vandenhende et al., 2021), recommender systems (Zhang et al., 2023a), and natural language processing (Chen et al., 2024b).

To bridge these gaps, this survey provides a dedicated, in-depth review of recent gradient-based MOO methods in supervised deep learning. Our main contributions are threefold:

- To the best of our knowledge, this is the first survey focused on gradient-based MOO methods in deep learning, addressing the limitations of previous literature that covers only a fraction of these algorithms.

- We propose a taxonomy that systematically categorizes gradient-based MOO methods based on their target output: finding a single solution, a finite set of solutions, or an infinite set of solutions. We further divide these into detailed subcategories, providing the first structured classification of its kind.

- Beyond a comprehensive algorithmic review, we provide an overview of the theoretical foundations, applications, resources, and future directions.

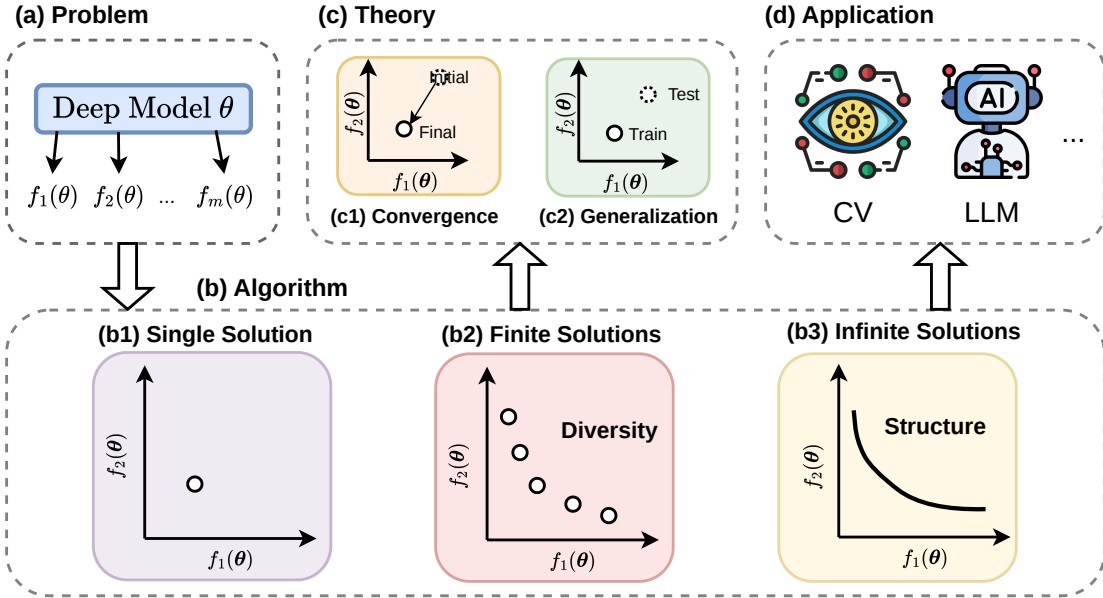

Figure 1: An overview of gradient-based multi-objective deep learning.

## 1.4    Organization of the Survey

In this survey, we categorize gradient-based multi-objective deep learning algorithms based on the type of output they obtain (Figure 2): (i) Algorithms that obtain a single, well-balanced model that is not overly biased toward any one objective: This is achieved through techniques like loss balancing and gradient balancing, which adjust the weight of each objective to resolve conflicts during optimization. (ii) Algorithms that obtain a finite set of Pareto-optimal solutions that allows users to select from multiple options based on their specific needs: This is accomplished by either dividing the problem into subproblems using preference vectors or directly optimizing for multiple solutions simultaneously. (iii) Algorithms that obtain a continuous set of Pareto-optimal models that can be generated on-demand for any given preference: This is achieved by learning a solution subspace using efficient structures, where preference vectors are randomly sampled during training and the optimization adapts techniques from single or finite solution methods.

The rest of this survey is organized as follows: Section 2 presents MOO preliminaries, including MOO definitions and concepts; Section 3 discusses methods for finding a single Pareto-optimal solution, covering loss balancing and gradient balancing approaches; Section 4 focuses on methods for identifying a set of finite Pareto-optimal solutions; Section 5 covers methods for identifying a set of infinite Pareto-optimal solutions; Section 6 delves into the theoretical analysis of convergence and generalization in gradient-based multi-objective optimization algorithms; Section 7 showcases various applications in deep learning, including computer vision, neural architecture search, recommender systems, and large language models; Section 8 offers useful resources on datasets and software tools for multi-objective deep learning; Section 9 highlights challenges in the field and suggests promising directions for future research; Section 10 summarizes this survey. An overview is shown in Figure 1.

## 1.5    Notations

The notations used in this survey are summarized in Table 1. Scalars are represented by non-bold letters; vectors and matrices are denoted by boldface type. Subscripts are used as indices for elements, and superscripts typically denote iteration numbers or indices within a solution set.

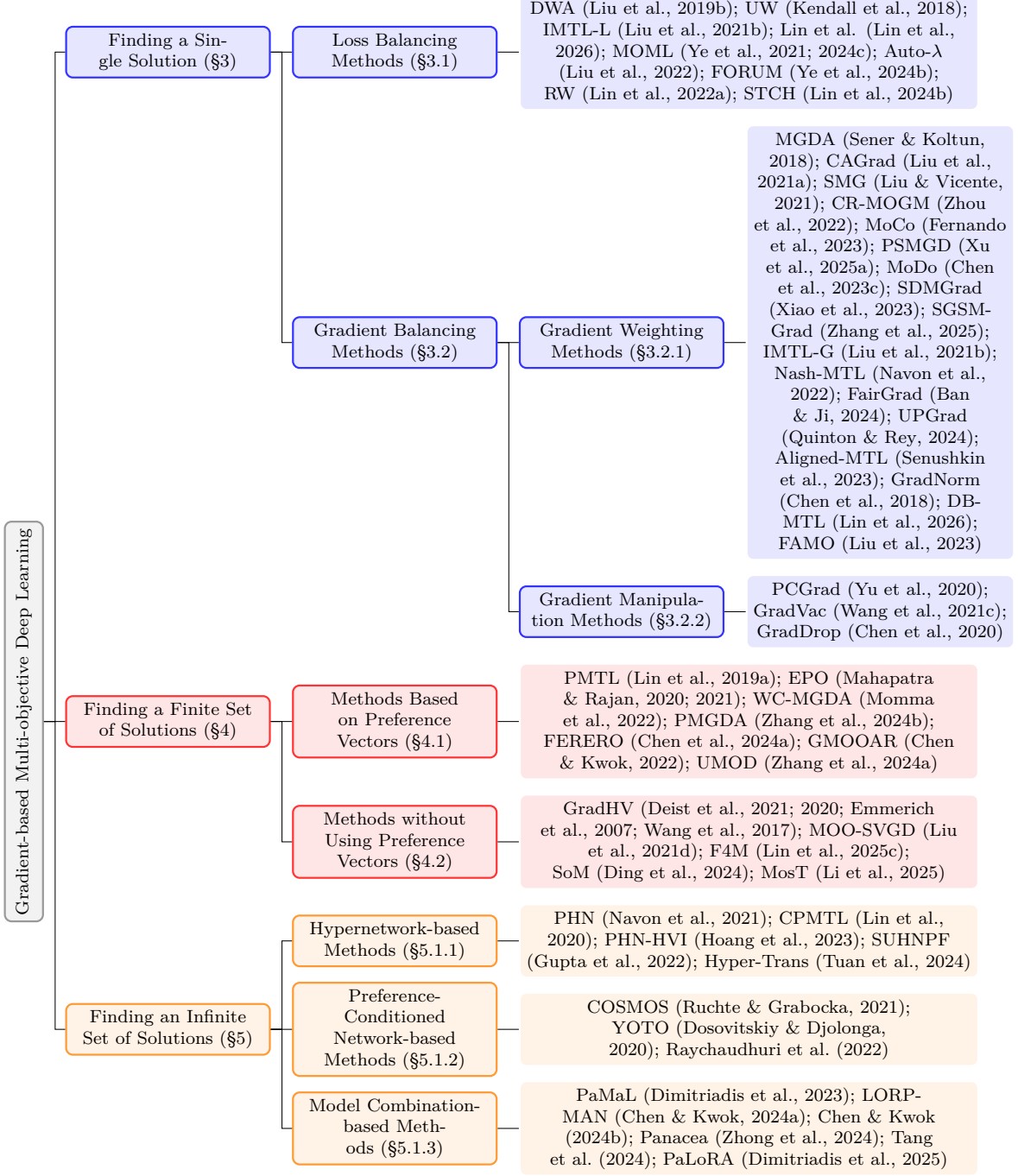

Figure 2: Taxonomy of existing gradient-based multi-objective deep learning algorithms.

## 2 Preliminary: Multi-Objective Optimization

In this section, we first introduce some MOO concepts in Section 2.1 and then review two classical methods (i.e., linear scalarization and Tchebycheff scalarization) for finding Pareto solutions of MOO problems in Section 2.2.

Table 1: Summary of notation.

| Notation | Description |
|---|---|
| $\boldsymbol{\theta} \in \mathcal{K} \subset \mathbb{R}^d$ | Decision variable $\boldsymbol{\theta}$ in feasible set $\mathcal{K}$ with dimension $d$; in deep learning, this usually corresponds to model parameters. |
| $m$ | Number of objectives. |
| $n$ | Number of finite Pareto solutions. |
| $K$ | Number of iterations. |
| $\boldsymbol{f} = [f_1, \ldots, f_m]^\top$ | Objective function. |
| $\boldsymbol{\lambda} = [\lambda_1, \ldots, \lambda_m]^\top$ | Objective weight vector. |
| $\boldsymbol{\alpha} = [\alpha_1, \ldots, \alpha_m]^\top$ | Preference vector. |
| $\boldsymbol{z}^* = [z_1^*, \ldots, z_m^*]^\top$ | Ideal objective value. |
| $\boldsymbol{\theta}^{(k)}$ | Solution at $k$-th iteration. |
| $\boldsymbol{\theta}^{(1)}, \ldots, \boldsymbol{\theta}^{(n)}$ | Solution 1 to Solution $n$ in a size-$n$ solution set. |
| $\boldsymbol{g}_i^{(k)}$ | The gradient vector of $i$-th objective at $k$-th iteration. |
| $\boldsymbol{G}^{(k)} = [\boldsymbol{g}_1^{(k)}, \ldots, \boldsymbol{g}_m^{(k)}]$ | Gradient matrix at $k$-th iteration. |
| $\boldsymbol{d}^{(k)}$ | Updating direction of $\boldsymbol{\theta}$ at $k$-th iteration. |
| $\phi$ | Parameters of Pareto set learning structures. |
| $\mathrm{HV}_{\boldsymbol{r}}(\cdot)$ | Hypervolume indicator with respect to reference point $\boldsymbol{r}$. |
| $[m]$ | Index set $\{1, \ldots, m\}$. |
| $\Delta_{m-1}$ | $(m-1)$-D simplex $\{\boldsymbol{\alpha} \mid \sum_{i=1}^m \alpha_i = 1, \alpha_i \geq 0, i \in [m]\}$. |
| $\|\cdot\|$ | $\ell_2$ norm. |
| $\eta$ | Step size for updating $\boldsymbol{\theta}$. |
| $\epsilon$ | Error tolerance. |

### 2.1 Key Concepts in Multi-Objective Optimization

Unlike single-objective optimization, solutions in MOO cannot be directly compared based on a single criterion, but are compared using the concept of dominance as follows. Note that without loss of generality, we consider multi-objective minimization (i.e., problem (1)) here.

**Definition 1** ((strict) Pareto dominance (Miettinen, 1999)). *A solution $\boldsymbol{\theta}^{(a)}$ dominates another solution $\boldsymbol{\theta}^{(b)}$ (denoted as $\boldsymbol{\theta}^{(a)} \preceq \boldsymbol{\theta}^{(b)}$) if and only if $f_i(\boldsymbol{\theta}^{(a)}) \leq f_i(\boldsymbol{\theta}^{(b)})$ for all $i \in [m]$, and there exists at least one $i \in [m]$ such that $f_i(\boldsymbol{\theta}^{(a)}) < f_i(\boldsymbol{\theta}^{(b)})$. A solution $\boldsymbol{\theta}^{(a)}$ strictly dominates another solution $\boldsymbol{\theta}^{(b)}$ if and only if $f_i(\boldsymbol{\theta}^{(a)}) < f_i(\boldsymbol{\theta}^{(b)})$ for all $i \in [m]$.*

Based on this definition, we further define Pareto-optimality (PO), Pareto set, and Pareto front as follows.

**Definition 2** ((weak) Pareto optimality (Miettinen, 1999)). *A solution $\boldsymbol{\theta}^*$ is Pareto-optimal if no other solution dominates it. A solution $\boldsymbol{\theta}^*$ is weakly Pareto-optimal if no other solution strictly dominates it.*

**Definition 3** (Pareto set (PS) and Pareto front (PF) (Miettinen, 1999)). *A PS is the set of all PO solutions. A PF is the set of all objective function values of the PO solutions.*

Figure 3 illustrates the Pareto concepts on the two-objective problem. The blue circles represent **Pareto-optimal solutions** (defined in Definition 2), which indicates that no solution can improve one objective without worsening the other. The blue curve connecting the blue circles denotes the **Pareto front**, as defined in Definition 3. The yellow circles indicate **weak Pareto-optimal solutions** since they can be improved in one objective without negatively impacting the other. For example, comparing $\boldsymbol{\theta}_2$ and $\boldsymbol{\theta}_1$, $\boldsymbol{\theta}_2$ is a weak Pareto-optimal solution since it can be improved in the second objective without affecting the first. The red circles represent solutions that are **not Pareto-optimal** because there exists another solution that strictly Pareto dominates them. For instance, comparing $\boldsymbol{\theta}_3$ and $\boldsymbol{\theta}_1$, $\boldsymbol{\theta}_3$ is not a Pareto-optimal solution as $\boldsymbol{\theta}_1$ outperforms it in both objectives.

**Definition 4** (Preference vector). *$\boldsymbol{\alpha}$ denotes a preference vector. The value of the entries of $\boldsymbol{\alpha}$ denote the preference of a specific objective. Throughout this paper, a preference vector $\boldsymbol{\alpha}$ is constrained on a*

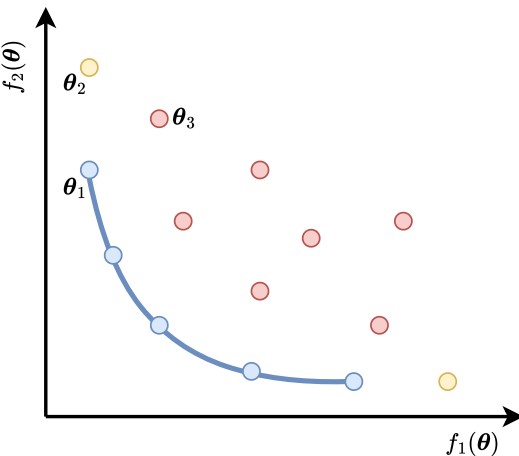

Figure 3: Illustration of Pareto optimality concepts. The blue, yellow, and red circles denote Pareto-optimal solutions, weakly Pareto-optimal solutions, and dominated solutions, respectively. The blue curve represents the Pareto front.

*simplex* $\Delta_{m-1}$, *where* $\Delta_{m-1} = \{\boldsymbol{\alpha} | \sum_{i=1}^{m} \alpha_i = 1, \alpha_i \geq 0, i \in [m]\}$. *The simplex constraint makes preferences nonnegative and normalized, so that $\boldsymbol{\alpha}$ represents relative trade-off importance rather than an arbitrary rescaling of the scalarized objective.*

In deep learning, the parameter $\boldsymbol{\theta}$ is typically optimized using gradient descent. In single-objective optimization, when the objective function is non-convex, the optimization process often reaches a stationary point. In the case of MOO, the solution generally converges to a *Pareto stationary* solution, which is defined as follows:

**Definition 5** (Pareto stationary (Désidéri, 2012)). *A solution $\boldsymbol{\theta}^*$ is called* Pareto stationary *if there exists a vector $\boldsymbol{\lambda} \in \Delta_{m-1}$ such that $\|\sum_{i=1}^{m} \lambda_i \nabla f_i(\boldsymbol{\theta}^*)\| = 0$.*

Pareto stationarity is a necessary condition for achieving Pareto optimality. If all objectives are convex with $\lambda_i > 0, \forall i$, it also serves as the Karush-Kuhn-Tucker (KKT) sufficient and necessary condition for Pareto optimality (Désidéri, 2012).

When evaluating the performance of a set of obtained solutions, the hypervolume indicator (HV) (Zitzler & Thiele, 1998) is one of the most widely used performance indicators, which is defined as follows.

**Definition 6** (Hypervolume indicator (Zitzler & Thiele, 1998)). *Given a solution set $\mathbb{S} = \{\boldsymbol{q}^{(1)}, \ldots, \boldsymbol{q}^{(N)}\}$ and a reference point $\boldsymbol{r}$, the hypervolume indicator of $\mathbb{S}$ is calculated by:*

$$\mathrm{HV}_{\boldsymbol{r}}(\mathbb{S}) = \Lambda(\boldsymbol{p} \mid \exists \boldsymbol{q} \in \mathbb{S} : \boldsymbol{q} \preceq \boldsymbol{p} \preceq \boldsymbol{r}), \tag{2}$$

*where $\Lambda(\cdot)$ denotes the Lebesgue measure of a set.*

HV provides a unary measure of both convergence and diversity of a solution set. In a two-objective minimization problem, it is the area dominated by the obtained objective vectors and bounded by the reference point; in higher dimensions, this area generalizes to a dominated volume. A larger HV indicates that the obtained solutions are more diverse and closer the PF in the objective space. In addition to HV, several other performance indicators are frequently utilized in MOO: (i) Generational Distance (GD) (Van Veldhuizen & Lamont, 1999): This metric measures the convergence of the obtained solution set by calculating the average Euclidean distance from each point in the solution set to the nearest point on the true PF. A smaller GD value indicates better convergence. (ii) Inverted Generational Distance (IGD) (Zitzler et al., 2003): Unlike GD, IGD measures the average distance from each point on the true PF to the nearest point in the obtained solution set. Consequently, IGD assesses both convergence and diversity simultaneously. A lower IGD value is desirable. For a more comprehensive review of performance indicators in MOO, we refer readers to (Jiang et al., 2014).

## 2.2 Scalarization Methods

**Linear Scalarization (LS).** The most straightforward way to solve MOO problems is to reformulate them as single-objective optimization problems weighted by the given preference vector $\boldsymbol{\alpha}$ as follows:

$$\min_{\boldsymbol{\theta}\in\mathcal{K}} g_{\boldsymbol{\alpha}}^{\text{LS}}(\boldsymbol{\theta}) := \sum_{i=1}^{m} \alpha_i f_i(\boldsymbol{\theta}). \tag{3}$$

LS is widely used in practice due to its simplicity. However, this method suffers from a limitation: it can only identify solutions on the convex part of the PF. For PFs with a concave shape, using LS only finds the endpoints of a PF.

**Tchebycheff Scalarization.** Another way to convert a MOO problem to a single objective optimization problem is to use the Tchebycheff scalarization function (Bowman Jr, 1976; Steuer & Choo, 1983), defined as:

$$\min_{\boldsymbol{\theta}\in\mathcal{K}} g_{\boldsymbol{\alpha}}^{\text{Tche}}(\boldsymbol{\theta}) := \max_{i\in[m]} \alpha_i(f_i(\boldsymbol{\theta}) - z_i^*), \tag{4}$$

where $z_i^*$ are reference values, usually set as the ideal value of the $i$-th objective. Unlike linear scalarization, Tchebycheff scalarization can explore the entire PF by traversing all preference vectors within the simplex. This ensures that both convex and concave regions of the PF are covered. The relationship between Tchebycheff scalarization and weak Pareto optimality can be formally described as follows:

**Theorem 1** ((Choo & Atkins, 1983))**.** *A solution $\boldsymbol{\theta}^*$ of the original MOO problem (1) is weakly Pareto-optimal if and only if there exists a preference vector $\boldsymbol{\alpha}$ such that $\boldsymbol{\theta}^*$ is an optimal solution of problem (4).*

If $\boldsymbol{\theta}^*$ is unique for a given $\boldsymbol{\alpha}$, it is considered Pareto-optimal. However, Tchebycheff scalarization poses challenges for gradient-based deep learning due to the nonsmooth nature of the $\max(\cdot)$ operator. This nonsmoothness leads to non-differentiability and a slow convergence rate of $\mathcal{O}(1/\epsilon^2)$, where $\epsilon$ represents the error tolerance (Lin et al., 2024b). Even when all objectives $\{f_i\}_{i=1}^{m}$ are differentiable, problem (4) is still non-differentiable.

# 3 Finding a Single Pareto-optimal Solution

In many real-world scenarios, such as multi-task learning (MTL) (Zhang & Yang, 2022; Ruder, 2017; Crawshaw, 2020), users genuinely require a single model that achieves a balance between objectives. We define "balance" as finding a fair compromise near the center of the Pareto front, ensuring that no single objective is disproportionately degraded in favor of others.

While using multiple task-specific networks offers flexibility, deploying a single balanced model is frequently necessary due to practical constraints. For example, in applications like autonomous driving, systems must perform multiple tasks simultaneously. Strict latency limits require these tasks to be executed in a single forward pass, making it impractical to run multiple models that require separate inferences. Similarly, in LLM alignment, a single generated response is evaluated simultaneously on multiple criteria. Since one output must satisfy all of these criteria at once, utilizing multiple separate models is not a viable solution.

In such cases, the goal is to identify a single, well-balanced Pareto-optimal solution. By default, objectives are treated as equally important, which corresponds to a uniform preference vector $\boldsymbol{\alpha} = [\frac{1}{m}, \ldots, \frac{1}{m}]^\top$. If the objectives carry unequal importance and a specific preference vector can be provided, users can simply rescale the objective functions accordingly. Once rescaled, the algorithms discussed in this section can then be applied to find the desired solution.

The most straightforward method is linear scalarization (i.e., problem (3)) with $\alpha_i = \frac{1}{m}$, which is known as Equal Weighting (EW) (Zhang & Yang, 2022) in MTL. However, EW may cause some tasks to have unsatisfactory performance (Standley et al., 2020; Lin et al., 2026). Therefore, to improve the performance, many methods have been proposed to dynamically tune the objective weights $\{\lambda_i\}_{i=1}^{m}$ during training, which

can in general be formulated as:

$$\min_{\boldsymbol{\theta}} \sum_{i=1}^{m} \lambda_i f_i(\boldsymbol{\theta}), \tag{5}$$

where $\lambda_i$ is the weight for the $i$-th objective $f_i(\boldsymbol{\theta})$ and the uniform preference vector $\boldsymbol{\alpha}$ is omitted for simplicity. These methods can be categorized as loss balancing methods and gradient balancing approaches. Some representative loss/gradient balancing methods are illustrated in Figure 4 (adapted from (Liu et al., 2021a;b)). A concise comparison of these methods along their type, qualitative per-iteration cost, and key idea is provided in Table 2.

## 3.1 Loss Balancing Methods

This type of approach dynamically computes or learns the objective weights $\{\lambda_i\}_{i=1}^{m}$ during training using some measures on loss such as the decrease speed of loss (Liu et al., 2019b), homeostatic uncertainty of loss (Kendall et al., 2018), loss scale (Liu et al., 2021b; Lin et al., 2026), and validation loss (Ye et al., 2021; 2024c;b; Liu et al., 2022).

Dynamic Weight Average (DWA) (Liu et al., 2019b) estimates each objective weight as the ratio of the training losses in the last two iterations. Formally, the objective weight in $k$-th iteration ($k > 2$) is computed as follows:

$$\lambda_i^{(k)} = \frac{m \exp(\omega_i^{(k-1)}/\gamma)}{\sum_{j=1}^{m} \exp(\omega_j^{(k-1)}/\gamma)}, \quad \omega_j^{(k-1)} = \frac{f_j^{(k-1)}}{f_j^{(k-2)}}, \tag{6}$$

where $\gamma$ is the temperature and $f_i^{(k)}$ is the loss value of $i$-th objective at $k$-th iteration.

Kendall et al. (2018) propose Uncertainty Weighting (UW) that considers the homoscedastic uncertainty of each objective and optimizes objective weights together with the model parameter as follows:

$$\min_{\boldsymbol{\theta}, \boldsymbol{s}} \sum_{i=1}^{m} \left( \frac{1}{2s_i^2} f_i(\boldsymbol{\theta}) + \log s_i \right), \tag{7}$$

where $\boldsymbol{s} = [s_1, \ldots, s_m]^\top$ is learnable and $\log s_i$'s are regularization terms.

Impartial Multi-Task Learning (IMTL) (Liu et al., 2021b) balances losses across tasks, abbreviated as IMTL-L(oss). Specifically, IMTL-L encourages all objectives to have a similar loss scale by transforming each objective $f_i(\boldsymbol{\theta})$ as $e^{s_i} f_i(\boldsymbol{\theta}) - s_i$. Similar to UW (Kendall et al., 2018), IMTL-L simultaneously learns the transformation parameter $\boldsymbol{s}$ and the model parameter $\boldsymbol{\theta}$ as follows:

$$\min_{\boldsymbol{\theta}, \boldsymbol{s}} \sum_{i=1}^{m} \left( e^{s_i} f_i(\boldsymbol{\theta}) - s_i \right). \tag{8}$$

Similar to IMTL-L (Liu et al., 2021b), Lin et al. (2026) also aim to make all objective losses have a similar scale. They achieve it by performing a logarithm transformation on each objective (i.e., $\log f_i(\boldsymbol{\theta})$). Moreover, they theoretically show that the transformation in IMTL-L is equivalent to the logarithm transformation when $s_i$ is the exact minimizer of $\min_{s_i} e^{s_i} f_i(\boldsymbol{\theta}) - s_i$ in each iteration. Hence, the logarithm transformation can recover the transformation in IMTL-L.

Ye et al. (2021; 2024c) propose Multi-Objective Meta Learning (MOML), which adaptively tunes the objective weights $\boldsymbol{\lambda}$ based on the validation performance, reformulating problem (5) as a multi-objective bi-level optimization problem:

$$\min_{\boldsymbol{\lambda}} \ \left[ f_1(\boldsymbol{\theta}^*(\boldsymbol{\lambda}); \mathcal{D}_1^{\text{val}}), \ldots, f_m(\boldsymbol{\theta}^*(\boldsymbol{\lambda}); \mathcal{D}_m^{\text{val}}) \right]^\top \tag{9a}$$

$$\text{s.t.} \ \ \boldsymbol{\theta}^*(\boldsymbol{\lambda}) = \arg\min_{\boldsymbol{\theta}} \sum_{i=1}^{m} \lambda_i f_i(\boldsymbol{\theta}; \mathcal{D}_i^{\text{tr}}), \tag{9b}$$

where $\mathcal{D}_i^{\mathrm{tr}}$ and $\mathcal{D}_i^{\mathrm{val}}$ are the training and validation datasets for the $i$-th objective, respectively. In each training iteration, when given objective weights $\{\lambda_i\}_{i=1}^m$, MOML first learns the model $\boldsymbol{\theta}^*(\boldsymbol{\lambda})$ on the training dataset by solving lower-level (LL) subproblem (9b) with $T$ iterations and then updates $\boldsymbol{\lambda}$ in the upper-level (UL) subproblem (9a) via minimizing the loss of the trained MTL model $\boldsymbol{\theta}^*(\boldsymbol{\lambda})$ on the validation dataset of each objective. However, when solving the UL subproblem (9a), MOML needs to calculate the complex hypergradient $\nabla_{\boldsymbol{\lambda}}\boldsymbol{\theta}^*(\boldsymbol{\lambda})$ which requires to compute many Hessian-vector products via the chain rule. Hence, the time and memory costs of MOML grow significantly fast with respect to the dimension of $\boldsymbol{\theta}$ and the number of LL iterations $T$.

Auto-$\lambda$ (Liu et al., 2022) modifies problem (9) by replacing the multi-objective UL subproblem (9a) with a single-objective problem: $\min_{\boldsymbol{\lambda}} \sum_{i=1}^m f_i(\boldsymbol{\theta}^*(\boldsymbol{\lambda}); \mathcal{D}_i^{\mathrm{val}})$. Besides, Auto-$\lambda$ approximates the hypergradient $\nabla_{\boldsymbol{\lambda}}\boldsymbol{\theta}^*(\boldsymbol{\lambda})$ via the finite difference approach. Thus, it is more efficient than MOML (Ye et al., 2021; 2024c).

FORUM (Ye et al., 2024b) is proposed to improve the efficiency of MOML (Ye et al., 2021; 2024c). FORUM first reformulates problem (9) as a constrained MOO problem via the value-function approach (Liu et al., 2021c) and then proposes a multi-gradient aggregation method to solve it. Compared with MOML, FORUM is a first-order method and does not need to compute the hypergradient $\nabla_{\boldsymbol{\lambda}}\boldsymbol{\theta}^*(\boldsymbol{\lambda})$. Thus, it is significantly more efficient than MOML.

Random Weighting (RW) (Lin et al., 2022a) randomly samples objective weights from a distribution (such as the standard normal distribution) and normalizes them into the simplex in each iteration. Lin et al. (2022a) report that RW obtains comparable performance with twelve MTL baselines on five image datasets and two multilingual tasks from XTREME.

Different from the above methods based on linear scalarization, Lin et al. (2024b) propose Smooth Tchebycheff scalarization (STCH), which transforms the Tchebycheff scalarization into a smooth function:

$$\min_{\boldsymbol{\theta}} f_{\boldsymbol{\alpha}}^{\mathrm{STCH}}(\boldsymbol{\theta}, \mu) := \mu \log \sum_{i=1}^m \exp\left(\frac{\alpha_i(f_i(\boldsymbol{\theta}) - z_i^*)}{\mu}\right), \tag{10}$$

where $\mu > 0$ is a smoothing parameter, and $\boldsymbol{z}^*$ is a reference point as in Tchebycheff scalarization (problem (4)). STCH improves upon the original Tchebycheff scalarization by making $f_{\boldsymbol{\alpha}}^{\mathrm{STCH}}(\boldsymbol{\theta}, \mu)$ smooth when all objective functions $f_i(\boldsymbol{\theta})$'s are smooth, resulting in faster convergence. It also retains the advantages of the original Tchebycheff scalarization: (1) convexity when all objective functions are convex and (2) Pareto optimality by appropriately choosing $\mu$.

## 3.2 Gradient Balancing Methods

Let $\boldsymbol{g}_i^{(k)} = \nabla_{\boldsymbol{\theta}} f_i(\boldsymbol{\theta})|_{\boldsymbol{\theta}^{(k)}} \in \mathbb{R}^d$ be the gradient of the $i$-th objective at the $k$-th iteration. This type of approach aims to find a common update direction $\boldsymbol{d}^{(k)}$ by adaptively aggregating the gradients of all objectives $\{\boldsymbol{g}_i^{(k)}\}_{i=1}^m$ at each iteration. Then, the model parameter is updated via $\boldsymbol{\theta}^{(k+1)} = \boldsymbol{\theta}^{(k)} - \eta\boldsymbol{d}^{(k)}$, where $\eta$ is a step size. To simplify notations, we omit the superscript in this section. Compared with loss balancing methods in Section 3.1, gradient balancing approaches offer more direct control over gradient conflict, although they usually require higher computational and memory costs.

### 3.2.1 Gradient Weighting Methods

In most gradient weighting methods, $\boldsymbol{d}$ is computed by:

$$\boldsymbol{d} = \boldsymbol{G}\boldsymbol{\lambda}, \tag{11}$$

where $\boldsymbol{G} = [\boldsymbol{g}_1, \ldots, \boldsymbol{g}_m] \in \mathbb{R}^{d \times m}$ is the gradient matrix and $\boldsymbol{\lambda} = [\lambda_1, \ldots, \lambda_m]^\top$ is the gradient weights.

Sener & Koltun (2018) apply Multiple-Gradient Descent Algorithm (MGDA) (Désidéri, 2012), which aims to find a direction $\boldsymbol{d}$ at each iteration to maximize the minimal decrease across the losses. This is formulated as:

$$\max_{\boldsymbol{d}} \min_{i \in [m]} (f_i(\boldsymbol{\theta}) - f_i(\boldsymbol{\theta} - \eta\boldsymbol{d})) \approx \max_{\boldsymbol{d}} \min_{i \in [m]} \boldsymbol{g}_i^\top \boldsymbol{d}, \tag{12}$$

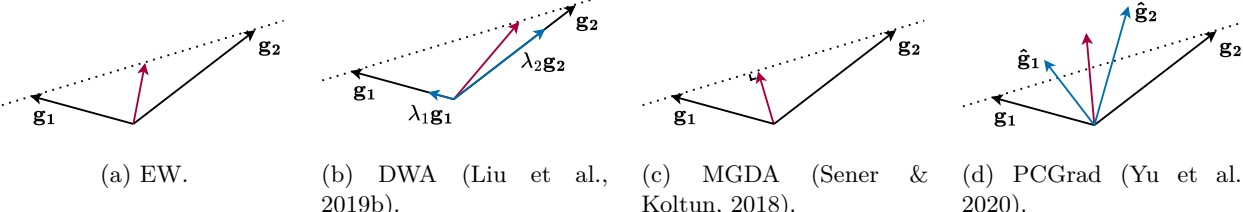

(a) EW.    (b) DWA (Liu et al., 2019b).    (c) MGDA (Sener & Koltun, 2018).    (d) PCGrad (Yu et al., 2020).

Figure 4: Illustration of the update direction $\boldsymbol{d}$ of several representative methods: EW, DWA (Liu et al., 2019b) from loss balancing methods, MGDA (Sener & Koltun, 2018) from gradient weighting methods, and PCGrad (Yu et al., 2020) from gradient manipulation methods. They are illustrated in a two-objective learning problem (two specific gradients are labeled as $\boldsymbol{g}_1$ and $\boldsymbol{g}_2$).

where $\eta$ is the step size. The solution of problem (12) is $\boldsymbol{d} = \boldsymbol{G}\boldsymbol{\lambda}$, where $\boldsymbol{\lambda}$ is computed as:

$$\boldsymbol{\lambda} = \underset{\boldsymbol{\lambda} \in \Delta_{m-1}}{\arg\min} \|\boldsymbol{G}\boldsymbol{\lambda}\|^2, \tag{13}$$

where $\Delta_{m-1}$ denotes the simplex. Sener & Koltun (2018) use the Frank-Wolfe algorithm (Jaggi, 2013) to solve problem (13).

Conflict-Averse Gradient descent (CAGrad) (Liu et al., 2021a) improves MGDA (Sener & Koltun, 2018) by constraining the aggregated gradient $\boldsymbol{d}$ to be around the average gradient $\boldsymbol{g}_0 = \frac{1}{m}\sum_{i=1}^m \boldsymbol{g}_i$ with a distance $c\|\boldsymbol{g}_0\|$, where $c \geq 0$ is a constant controlling the conflict-aversion strength. Specifically, $\boldsymbol{d}$ is computed by solving the following problem:

$$\max_{\boldsymbol{d}} \min_{i \in [m]} \boldsymbol{g}_i^\top \boldsymbol{d}, \quad \text{s.t.} \quad \|\boldsymbol{d} - \boldsymbol{g}_0\| \leq c\|\boldsymbol{g}_0\|. \tag{14}$$

Problem (14) is equivalent to first computing $\boldsymbol{\lambda}$ by solving the following problem:

$$\boldsymbol{\lambda} = \underset{\boldsymbol{\lambda} \in \Delta_{m-1}}{\arg\min} \boldsymbol{g}_{\boldsymbol{\lambda}}^\top \boldsymbol{g}_0 + c\|\boldsymbol{g}_0\|\|\boldsymbol{g}_{\boldsymbol{\lambda}}\|, \tag{15}$$

where $\boldsymbol{g}_{\boldsymbol{\lambda}} = \frac{1}{m}\boldsymbol{G}\boldsymbol{\lambda}$ and then calculate the update direction $\boldsymbol{d} = \boldsymbol{g}_0 + \frac{c\|\boldsymbol{g}_0\|}{\|\boldsymbol{g}_{\boldsymbol{\lambda}}\|}\boldsymbol{g}_{\boldsymbol{\lambda}}$. Note that CAGrad is simplified to EW when $c = 0$ and degenerates into MGDA (Sener & Koltun, 2018) when $c \to +\infty$.

MGDA (Sener & Koltun, 2018) and its variant CAGrad (Liu et al., 2021a) can converge to a Pareto stationary point in the deterministic setting with full-gradient computations. However, in deep learning, where stochastic gradients are used, many works (Liu & Vicente, 2021; Zhou et al., 2022; Xu et al., 2025a; Fernando et al., 2023; Chen et al., 2023c; Xiao et al., 2023; Zhang et al., 2025) explore the convergence properties of MGDA under stochastic gradients, which are introduced in detail in Section 6.1.

IMTL-G(rad) (Liu et al., 2021b) aims to find $\boldsymbol{d}$ that has equal projections on all objective gradients. Thus, we have:

$$\boldsymbol{u}_1^\top \boldsymbol{d} = \boldsymbol{u}_i^\top \boldsymbol{d}, \quad 2 \leq i \leq m, \tag{16}$$

where $\boldsymbol{u}_i = \boldsymbol{g}_i/\|\boldsymbol{g}_i\|$. If constraining $\sum_{i=1}^m \lambda_i = 1$, problem (16) has a closed-form solution of $\boldsymbol{\lambda}$:

$$\boldsymbol{\lambda}_{(2,\ldots,m)} = \boldsymbol{g}_1^\top \boldsymbol{U}\left(\boldsymbol{D}\boldsymbol{U}^\top\right)^{-1}, \quad \lambda_1 = 1 - \sum_{i=2}^m \lambda_i, \tag{17}$$

where $\boldsymbol{\lambda}_{(2,\ldots,m)} = [\lambda_2, \ldots, \lambda_m]^\top$, $\boldsymbol{U} = [\boldsymbol{u}_1 - \boldsymbol{u}_2, \ldots, \boldsymbol{u}_1 - \boldsymbol{u}_m]$, and $\boldsymbol{D} = [\boldsymbol{g}_1 - \boldsymbol{g}_2, \ldots, \boldsymbol{g}_1 - \boldsymbol{g}_m]$.

Nash-MTL (Navon et al., 2022) formulates problem (11) as a bargaining game (Nash, 1953; Thomson, 1994), where each objective is cast as a player and the utility function for each player is defined as $\boldsymbol{g}_i^\top \boldsymbol{d}$. Hence, $\boldsymbol{d}$

is computed by solving the following problem:

$$\max_{\boldsymbol{d}} \sum_{i=1}^{m} \log\left(\boldsymbol{g}_i^\top \boldsymbol{d}\right). \tag{18}$$

The solution of problem (18) is $\boldsymbol{d} = \boldsymbol{G}\boldsymbol{\lambda}$, where $\boldsymbol{\lambda}$ is obtained by solving the following problem:

$$\boldsymbol{G}^\top \boldsymbol{G} \boldsymbol{\lambda} = \boldsymbol{\lambda}^{-1}, \tag{19}$$

which can be approximately solved using a sequence of convex optimization problems.

FairGrad (Ban & Ji, 2024) extends Nash-MTL (Navon et al., 2022) by computing $\boldsymbol{\lambda}$ through the equation $\boldsymbol{G}^\top \boldsymbol{G} \boldsymbol{\lambda} = \boldsymbol{\lambda}^{-1/\gamma}$, where $\gamma \geq 0$ is a constant. Unlike Nash-MTL (Navon et al., 2022), FairGrad treats the problem as a constrained nonlinear least squares problem. Similarly, UPGrad (Quinton & Rey, 2024) calculates $\boldsymbol{\lambda}$ by optimizing $\min_{\boldsymbol{\lambda}} \boldsymbol{\lambda}^\top (\boldsymbol{G}^\top \boldsymbol{G}) \boldsymbol{\lambda}$, a quadratic programming (QP) problem.

Aligned-MTL (Senushkin et al., 2023) considers problem (11) as a linear system and aims to enhance its stability by minimizing the condition number of $\boldsymbol{G}$. Specifically, the Gram matrix $\boldsymbol{C} = \boldsymbol{G}^\top \boldsymbol{G}$ is first computed. Then, the eigenvalues $\{\sigma_1, \ldots, \sigma_R\}$ and eigenvectors $\boldsymbol{V}$ of $\boldsymbol{C}$ are obtained. Finally, $\boldsymbol{\lambda}$ is computed by:

$$\boldsymbol{\lambda} = \sqrt{\sigma_R} \boldsymbol{V} \boldsymbol{\Sigma}^{-1} \boldsymbol{V}^\top, \tag{20}$$

where $\boldsymbol{\Sigma}^{-1} = \mathrm{diag}\left(\sqrt{1/\sigma_1}, \ldots, \sqrt{1/\sigma_R}\right)$.

Gradient Normalization (GradNorm) (Chen et al., 2018) aims to learn $\boldsymbol{\lambda}$ so that the scaled gradients have similar magnitudes. It updates $\boldsymbol{\lambda}$ by solving the following problem via one-step gradient descent:

$$\min_{\boldsymbol{\lambda}} \sum_{i=1}^{m} \left(\|\lambda_i \boldsymbol{g}_i\| - c \times r_i^\gamma\right), \tag{21}$$

where $c = \frac{1}{m} \sum_{i=1}^{m} \|\lambda_i \boldsymbol{g}_i\|$ is the average scaled gradient norm and is treated as a constant, $r_i = \frac{f_i/f_i^{(0)}}{\frac{1}{m} \sum_{j=1}^{m} (f_j/f_j^{(0)})}$ is the training speed of the $i$-th objective and is also considered as a constant, and $\gamma > 0$ is a hyperparameter.

Dual-Balancing Multi-Task Learning (DB-MTL) (Lin et al., 2026) also normalizes all objective gradients to the same magnitude. $\boldsymbol{\lambda}$ is computed as $\lambda_i = \gamma/\|\boldsymbol{g}_i\|$, where $\gamma = \max_{i \in [m]} \|\boldsymbol{g}_i\|$ is a scaling factor controlling the update magnitude. Thus, compared with GradNorm (Chen et al., 2018), this method guarantees all objective gradients have the same norm in each iteration. Moreover, they observe that the choice of the update magnitude $\gamma$ significantly affects performance.

### 3.2.2 Gradient Manipulation Methods

To address gradient conflicts, gradient manipulation methods correct each objective gradient $\boldsymbol{g}_i$ to $\hat{\boldsymbol{g}}_i$ and then compute the update direction as $\boldsymbol{d} = \sum_{i=1}^{m} \hat{\boldsymbol{g}}_i$.

Projecting Conflicting Gradients (PCGrad) (Yu et al., 2020) considers two objective gradients $\boldsymbol{g}_i$ and $\boldsymbol{g}_j$ as conflicting if $\boldsymbol{g}_i^\top \boldsymbol{g}_j < 0$. Then, it corrects each objective gradient by projecting it onto the normal plane of other objectives' gradients. Specifically, at each iteration, $\hat{\boldsymbol{g}}_i$ is initialized with its original gradient $\boldsymbol{g}_i$. Then, for each $j \neq i$, if $\hat{\boldsymbol{g}}_i^\top \boldsymbol{g}_j < 0$, $\hat{\boldsymbol{g}}_i$ is corrected as:

$$\hat{\boldsymbol{g}}_i = \hat{\boldsymbol{g}}_i - \frac{\hat{\boldsymbol{g}}_i^\top \boldsymbol{g}_j}{\|\boldsymbol{g}_j\|^2} \boldsymbol{g}_j. \tag{22}$$

This reduces gradient conflicts by removing the components of $\hat{\boldsymbol{g}}_i$ that oppose other objective gradients.

While PCGrad (Yu et al., 2020) corrects the gradient if and only if two objectives have a negative gradient similarity, Gradient Vaccine (GradVac) (Wang et al., 2021c) extends it to a more general and adaptive formulation. In each iteration, if $\phi_{ij} < \bar{\phi}_{ij}$, GradVac updates the corrected gradient $\hat{\boldsymbol{g}}_i$ as:

$$\hat{\boldsymbol{g}}_i = \hat{\boldsymbol{g}}_i - \frac{\|\hat{\boldsymbol{g}}_i\| \left( \bar{\phi}_{ij}\sqrt{1-\phi_{ij}^2} - \phi_{ij}\sqrt{1-\bar{\phi}_{ij}^2} \right)}{\|\boldsymbol{g}_j\| \sqrt{1-\bar{\phi}_{ij}^2}} \boldsymbol{g}_j, \tag{23}$$

where $\phi_{ij}$ is the cosine similarity between $\hat{\boldsymbol{g}}_i$ and $\boldsymbol{g}_j$. $\bar{\phi}_{ij}$ is initialized to 0 and updated by Exponential Moving Average (EMA), i.e., $\bar{\phi}_{ij} \leftarrow (1-\beta)\bar{\phi}_{ij} + \beta\phi_{ij}$, where $\beta$ is a hyperparameter. Note that GradVac simplifies to PCGrad when $\bar{\phi}_{ij} = 0$. Gradient Sign Dropout (GradDrop) (Chen et al., 2020) considers that conflicts arise from differences in the sign of gradient values. Thus, a probabilistic masking procedure is proposed to preserve gradients with consistent signs during each update.

### 3.2.3 Practical Speedup Strategy

Gradient balancing methods suffer from high computational and storage costs. Specifically, in each iteration, almost all gradient balancing methods require performing $m$ back-propagation processes to obtain all objective gradients w.r.t. the model parameter $\boldsymbol{\theta} \in \mathbb{R}^d$ and then store these gradients $\boldsymbol{G} \in \mathbb{R}^{d \times m}$. This can be computationally expensive when dealing with a large number of objectives or using a neural network with a large number of parameters. Moreover, many gradient balancing methods (such as MGDA (Sener & Koltun, 2018), CAGrad (Liu et al., 2021a), Nash-MTL (Navon et al., 2022), and FairGrad (Ban & Ji, 2024)) need to solve an optimization problem to obtain the objective weight $\boldsymbol{\lambda}$ in each iteration, which also increases the computational and memory costs. Hence, several strategies are proposed to alleviate this problem.

Sener & Koltun (2018) use feature-level gradients (i.e., gradients w.r.t. the representations from the last shared layer) to replace the parameter-level gradients (i.e., gradients w.r.t. the shared parameters $\boldsymbol{\theta}$). Since the dimension of the representation is much smaller than that of the shared parameters, it can significantly reduce the computational and memory costs. This approximation is used to compute the objective weights or update direction; the shared parameters are still updated by backpropagating the resulting scalarized or aggregated loss through the network. This strategy is also adopted by some gradient balancing methods, such as IMTL-G (Liu et al., 2021b) and Aligned-MTL (Senushkin et al., 2023).

Liu et al. (2021a) randomly select a subset of objectives to calculate the update direction in each iteration. Navon et al. (2022) propose to update the objective weight $\boldsymbol{\lambda}$ every $\tau$ iterations instead of updating in each iteration. Although this strategy speeds up training, they observe that it may cause a noticeable drop in performance. Liu et al. (2023) consider optimizing the logarithm of the MGDA objective (Sener & Koltun, 2018) and propose a speedup strategy. Specifically, when solving problem (13) via one-step gradient descent, $\boldsymbol{\lambda}$ is updated as $\boldsymbol{\lambda} \leftarrow \boldsymbol{\lambda} - \eta\nabla_{\boldsymbol{\lambda}} \|\boldsymbol{G}\boldsymbol{\lambda}\|^2$. Note that

$$\frac{1}{2}\nabla_{\boldsymbol{\lambda}} \|\boldsymbol{G}\boldsymbol{\lambda}\|^2 = \boldsymbol{G}^\top\boldsymbol{G}\boldsymbol{\lambda} = \boldsymbol{G}^\top\boldsymbol{d} \approx \frac{1}{\eta}\left[ f_1^{(k)} - f_1^{(k+1)}, \dots, f_m^{(k)} - f_m^{(k+1)} \right]^\top, \tag{24}$$

where $f_i^{(k)}$ is the loss value of the $i$-th objective in the $k$-th iteration. Hence, $\boldsymbol{\lambda}$ can be approximately updated using the change in losses without computing all objective gradients. Although this strategy significantly reduces computational and memory costs, it may cause performance degradation. Moreover, it is only applicable to MGDA-based methods.

### 3.3 Discussions

**Computational complexity.** Table 2 summarizes representative methods by type, qualitative per-iteration cost, and key idea. Let $d$ be the number of parameters over which the objective gradients are aggregated (the shared parameters in multi-task architectures with task-specific heads, or the entire model when all objectives share parameters, as in LLM alignment), $m$ the number of objectives, and $C_{\mathrm{bp}}$ the cost of one backward pass. Each iteration also includes a forward pass with cost $\mathcal{O}(C_{\mathrm{bp}})$. This base cost is common to all methods, so the main per-iteration difference comes from backward passes and auxiliary optimization.

Table 2: Representative methods for finding a single Pareto-optimal solution.

| Method | Type | Cost | Key idea |
|---|---|---|---|
| DWA (Liu et al., 2019b) | Loss balancing | Low | weights from the rate of loss decrease |
| UW (Kendall et al., 2018) | Loss balancing | Low | homoscedastic-uncertainty weighting |
| IMTL-L (Liu et al., 2021b) | Loss balancing | Low | enforces equal loss scale |
| Log-transform (Lin et al., 2026) | Loss balancing | Low | logarithmic loss scaling |
| RW (Lin et al., 2022a) | Loss balancing | Low | random objective weights |
| STCH (Lin et al., 2024b) | Loss balancing | Low | smooth Tchebycheff scalarization |
| MOML (Ye et al., 2021; 2024c) | Loss balancing | Medium | bi-level weighting via validation loss (multi-objective) |
| Auto-$\lambda$ (Liu et al., 2022) | Loss balancing | Medium | bi-level weighting via validation loss (scalarized) |
| FORUM (Ye et al., 2024b) | Loss balancing | Medium | first-order bi-level weighting |
| MGDA (Sener & Koltun, 2018) | Gradient weighting | High | minimum-norm common descent direction |
| CAGrad (Liu et al., 2021a) | Gradient weighting | High | conflict-averse direction near the average gradient |
| Nash-MTL (Navon et al., 2022) | Gradient weighting | High | bargaining-game weighting |
| FairGrad (Ban & Ji, 2024) | Gradient weighting | High | fairness-driven weight allocation |
| UPGrad (Quinton & Rey, 2024) | Gradient weighting | High | projection-based update direction |
| IMTL-G (Liu et al., 2021b) | Gradient weighting | High | equal gradient projections |
| Aligned-MTL (Senushkin et al., 2023) | Gradient weighting | High | minimizes the gradient condition number |
| GradNorm (Chen et al., 2018) | Gradient weighting | High | equalizes scaled gradient norms |
| DB-MTL (Lin et al., 2026) | Gradient weighting | High | equal gradient norms |
| FAMO (Liu et al., 2023) | Gradient weighting | Low | loss-based approximation of MGDA |
| PCGrad (Yu et al., 2020) | Gradient manipulation | High | projects out conflicting gradient components |
| GradVac (Wang et al., 2021c) | Gradient manipulation | High | adaptive gradient projection |
| GradDrop (Chen et al., 2020) | Gradient manipulation | High | sign-based gradient masking |

Most loss balancing methods need a single backward pass and thus have essentially the same per-iteration cost as EW, namely $\mathcal{O}(C_{\mathrm{bp}})$; the bi-level variants MOML (Ye et al., 2021; 2024c), Auto-$\lambda$ (Liu et al., 2022), and FORUM (Ye et al., 2024b) are more expensive because they additionally differentiate through an inner optimization. Gradient balancing typically computes the gradient matrix $G \in \mathbb{R}^{d \times m}$, costing $\mathcal{O}(mC_{\mathrm{bp}})$ time and $\mathcal{O}(md)$ memory; many gradient weighting methods then form the Gram matrix $G^\top G$ at a cost of $\mathcal{O}(m^2 d)$ before solving a small $m$-dimensional subproblem. The dominant overhead is usually the $m$ backward passes, while the Gram-matrix and solver costs become more visible when the number of objectives is large. Feature-level gradients, periodic updates, objective subsampling, and FAMO (Liu et al., 2023) mainly target this overhead.

**Strengths, weaknesses, and recommendations.** In practice, loss balancing is the cheapest default and is easy to add to existing training code, but it controls conflict only indirectly through loss statistics. Gradient weighting and manipulation methods are preferable when gradient conflict is severe or Pareto-stationarity is the main optimization target, but their $m$-pass cost can be prohibitive for many objectives or very large models. Closed-form methods such as IMTL-G and DB-MTL are simple to deploy, while MGDA, CAGrad, and Nash-MTL remain common solver-based baselines. Since no method dominates across benchmarks, LibMTL (Lin & Zhang, 2023) (Section 8) is useful for controlled comparison. All methods in this section return only one solution, so Sections 4 and 5 are needed when users must inspect multiple trade-offs.

## 4 Finding a Finite Set of Solutions

In some scenarios, a single solution may be insufficient to balance the objectives, as users may prefer to obtain multiple trade-off solutions and select one based on their specific needs. This section introduces algorithms to identify a finite set of solutions, providing a discrete approximation of the Pareto set. We first discuss methods based on preference vectors (Section 4.1), followed by approaches that do not require the use of preference vectors (Section 4.2). Figure 5 shows solutions obtained by some representative methods on LSMOP1 (Cheng et al., 2016) after 5000 iterations of each algorithm. Table 3 further compares these

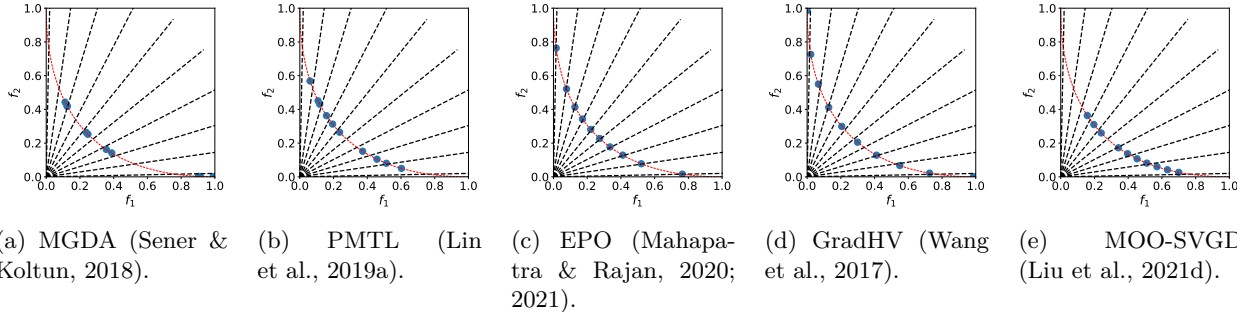

(a) MGDA (Sener & Koltun, 2018).   (b) PMTL (Lin et al., 2019a).   (c) EPO (Mahapatra & Rajan, 2020; 2021).   (d) GradHV (Wang et al., 2017).   (e) MOO-SVGD (Liu et al., 2021d).

Figure 5: Distributions of solutions obtained by using representative methods: PMTL (Lin et al., 2019a) and EPO (Mahapatra & Rajan, 2020; 2021) are preference vector-based methods, while GradHV (Wang et al., 2017) and MOO-SVGD (Liu et al., 2021d) are methods not using preference vectors. We also include MGDA (Sener & Koltun, 2018) as a reference; the multiple MGDA points are obtained from independent MGDA runs with different random model-parameter initializations, since MGDA itself is a single-solution method and has no mechanism to promote diversity. The blue circles denote the solutions, the red curves denote the ground truth PF and the black lines denote the preference vectors. The figure shows that MGDA (Sener & Koltun, 2018) has poor diversity because it lacks preference incorporation or diversity promotion. PMTL (Lin et al., 2019a) improves alignment with the preference vectors but still falls short of exact alignment, while EPO (Mahapatra & Rajan, 2020; 2021) achieves exact alignment. GradHV (Wang et al., 2017) and MOO-SVGD (Liu et al., 2021d) can also generate diverse solutions although they do not use preference vectors.

methods in terms of preference handling, alignment exactness, and whether the solutions are optimized jointly.

## 4.1 Methods Based on Preference Vectors

Preference vector-based methods rely on a preference vector set $\{\boldsymbol{\alpha}^{(1)}, \ldots, \boldsymbol{\alpha}^{(n)}\}$. These preference vectors partition the problem into $n$ subproblems, where each subproblem involves finding the solution that corresponds to a specific preference vector in the set. By solving $n$ subproblems, these methods obtain a set of $n$ solutions.

Pareto Multi-Task Learning (PMTL) (Lin et al., 2019a), inspired by the idea of decomposition-based MOO algorithms (Zhang & Li, 2007), incorporates preference vectors as constraints on the objectives. For the $i$-th subproblem with preference vector $\boldsymbol{\alpha}^{(i)}$, the objective vector $\boldsymbol{f}(\boldsymbol{\theta})$ is constrained to be closer to $\boldsymbol{\alpha}^{(i)}$ than to the other preference vectors:

$$\min_{\boldsymbol{\theta} \in \mathcal{K} \subset \mathbb{R}^d} \boldsymbol{f}(\boldsymbol{\theta}) := [f_1(\boldsymbol{\theta}), \ldots, f_m(\boldsymbol{\theta})]^\top,$$
$$\text{s.t. } r^{(j)}(\boldsymbol{\theta}) := (\boldsymbol{\alpha}^{(j)} - \boldsymbol{\alpha}^{(i)})^\top \boldsymbol{f}(\boldsymbol{\theta}) \leq 0, \quad j \in [n]. \tag{25}$$

Let $\boldsymbol{R}$ be the matrix containing gradients of the active constraints, i.e, $\boldsymbol{R} = \left[\nabla r^{(j)}(\boldsymbol{\theta})\right]_{j \in \mathbb{I}(\boldsymbol{\theta})}$, with $\mathbb{I}(\boldsymbol{\theta}) = \{j \in [n] \mid r^{(j)}(\boldsymbol{\theta}) \geq -\epsilon\}$ containing indices of the active constraints. Using a method akin to MGDA (Sener & Koltun, 2018), PMTL derives a descent direction for problem (25) that optimizes all $m$ objectives while keeping $\boldsymbol{f}(\boldsymbol{\theta})$ within the sector closer to the preference vector $\boldsymbol{\alpha}^{(i)}$. The descent direction is:

$$\boldsymbol{d} = -\boldsymbol{G}\boldsymbol{\lambda} + \boldsymbol{R}\boldsymbol{\beta}, \tag{26}$$

where $\boldsymbol{G} = [\nabla f_1(\boldsymbol{\theta}), \ldots, \nabla f_m(\boldsymbol{\theta})]$ is the Jacobian matrix. $\boldsymbol{\lambda}$ and $\boldsymbol{\beta}$ are coefficients obtained by solving the following quadratic programming problem:

$$\min_{\boldsymbol{\lambda}, \boldsymbol{\beta}} \|\boldsymbol{G}\boldsymbol{\lambda} + \boldsymbol{R}\boldsymbol{\beta}\|^2, \quad \text{s.t. } \boldsymbol{\lambda}^\top \mathbf{1} + \boldsymbol{\beta}^\top \mathbf{1} = 1, \lambda_i \geq 0, \beta_j \geq 0. \tag{27}$$

A limitation of PMTL is it can only constrain objective vectors within sectors, which lacks precise control over the position of Pareto solutions.

Exact Pareto Optimal search (EPO) (Mahapatra & Rajan, 2020; 2021) is designed to locate Pareto solutions which are aligned the objective vector exactly with given preference vectors. EPO achieves this using a uniformity function, $u_{\boldsymbol{\alpha}}(\boldsymbol{f}(\boldsymbol{\theta})) = \mathrm{KL}(\hat{\boldsymbol{f}}(\boldsymbol{\theta}) \parallel \mathbb{1}/m)$, where: $\hat{f}_i(\boldsymbol{\theta}) = \frac{\alpha_i f_i(\boldsymbol{\theta})}{\sum_{j=1}^{m} \alpha_j f_j(\boldsymbol{\theta})}$. Minimizing this function aligns $\boldsymbol{f}(\boldsymbol{\theta})$ with $\boldsymbol{\alpha}$, achieving precise alignment and Pareto optimality. Let $\boldsymbol{C} = \boldsymbol{G}^{\top}\boldsymbol{G}$ and $\boldsymbol{c}_i$ be the $i$-th column of $\boldsymbol{C}$. Let $\boldsymbol{a} = [a_1, \ldots, a_m]^{\top}$, where $a_i = \alpha_i(\log(m\hat{f}_i(\boldsymbol{\theta})) - u_{\boldsymbol{\alpha}}(f(\boldsymbol{\theta})))$. At each iteration, EPO classifies objective indices into three sets: $\mathbb{J} = \{j \mid \boldsymbol{a}^{\top}\boldsymbol{c}_j > 0\}$ are the indices decreasing uniformity, $\overline{\mathbb{J}} = \{j \mid \boldsymbol{a}^{\top}\boldsymbol{c}_j \leq 0\}$ are the indices increasing uniformity, and $\mathbb{J}^*$ are the indices with the maximum $\alpha_j f_j(\boldsymbol{\theta})$. Based on these sets, it determines $\boldsymbol{\lambda}$ for the common descent direction $\boldsymbol{d} = \boldsymbol{G}\boldsymbol{\lambda}$ by solving:

$$\max_{\boldsymbol{\lambda} \in \Delta_{m-1}} \boldsymbol{\lambda}^{\top}\boldsymbol{C}(\boldsymbol{a}\mathbb{1}_{u_{\boldsymbol{\alpha}}} + \mathbf{1}(1 - \mathbb{1}_{u_{\boldsymbol{\alpha}}})), \quad \text{s.t.} \left\{ \begin{array}{l} \boldsymbol{\lambda}^{\top}\boldsymbol{c}_j \geq \boldsymbol{a}^{\top}\boldsymbol{c}_j\mathbb{1}_{\mathbb{J}}, \quad \forall j \in \overline{\mathbb{J}} \setminus \mathbb{J}^*, \\ \boldsymbol{\lambda}^{\top}\boldsymbol{c}_j \geq 0, \quad \forall j \in \mathbb{J}^*. \end{array} \right. \tag{28}$$

This updating direction can balance objectives while reducing non-uniformity, resulting in solutions that exactly match each preference vector. However, EPO has three drawbacks: (1) objective vectors must be non-negative, (2) unnecessary complexity from dividing sets into three subsets, and (3) computational inefficiency due to repeated Jacobian calculations and solving linear programming problem.

To address these drawbacks, Weighted Chebyshev MGDA (WC-MGDA) (Momma et al., 2022) considers the dual form of Tchebycheff scalarization (problem (4)) of a linear programming problem. Optimizing Tchebycheff function addresses the previously mentioned drawbacks of EPO. However, its convergence rate is relatively slow. Two methods for improving the Tchebycheff scalarization include smooth Tchebycheff scalarization (Lin et al., 2024b), which replaces the non-smooth max operation with a smooth approximation, and Preference-based MGDA (PMGDA) (Zhang et al., 2024b), which only requires the update direction to have a negative inner product with the exact constraint gradient.

FERERO (Chen et al., 2024a) captures preference information within the MGDA framework, addressing both objective constraints with constants and the exactness constraint. It achieves fast convergence rates of $\mathcal{O}(\epsilon^{-1})$ for deterministic gradients and $\mathcal{O}(\epsilon^{-2})$ for stochastic gradients, where $\epsilon$ is the error tolerance.

The methods discussed above rely on a fixed set of preference vectors. However, in real-world applications where the shape of the Pareto front is unknown, a predefined set of preference vectors may not always result in well-distributed solutions. To overcome this limitation, GMOOAR (Chen & Kwok, 2022) formulates the problem as a bi-level optimization problem. In the upper level, preference vectors are optimized to maximize either the hypervolume or uniformity of the solutions. In the lower level, solutions are optimized based on these preference vectors. This approach enables the algorithm to dynamically adjust preference vectors based on the given optimization problem, ensuring the desired solution distribution. UMOD (Zhang et al., 2024a) introduces an approach that aims to maximize the minimum distance between solutions:

$$\max_{\boldsymbol{\theta}^{(i)}, \boldsymbol{\theta}^{(j)} \in \mathrm{PS}} \min_{1 \leq i < j \leq n} \rho(\boldsymbol{f}(\boldsymbol{\theta}^{(i)}), \boldsymbol{f}(\boldsymbol{\theta}^{(j)})), \tag{29}$$

where $\rho(\cdot, \cdot)$ denotes the Euclidean distance. UMOD achieves desirable solution distributions with theoretical guarantees: (1) For bi-objective problems with a connected, compact PF, the optimal solution includes the PF endpoints, with equal distances between neighboring objective vectors; (2) As the number of solutions increases, the objective vectors asymptotically distribute on the PF.

### 4.2 Methods without Using Preference Vectors

Unlike methods that rely on preference vectors, another approach directly optimizes for a diverse set of Pareto-optimal solutions. As HV evaluates solution sets based on both convergence and diversity, it is commonly used in traditional evolutionary MOO (Shang et al., 2020). In the context of gradient-based MOO, gradient-based hypervolume maximization algorithms (GradHV) (Wang et al., 2017; Deist et al., 2021; 2020; Emmerich et al., 2007) have been developed. Let $\{\boldsymbol{\theta}^{(1)}, \ldots, \boldsymbol{\theta}^{(n)}\}$ be a set of solutions and

$\{\boldsymbol{f}(\boldsymbol{\theta}^{(1)}), \ldots, \boldsymbol{f}(\boldsymbol{\theta}^{(n)})\}$ be their corresponding objective vectors. These algorithms first calculate the hyper-volume gradient of the $i$-th solution $\boldsymbol{\theta}^{(i)}$ as follows:

$$\boldsymbol{d}^{(i)} = \sum_{k=1}^{m} \underbrace{\frac{\partial \mathrm{HV}_{\boldsymbol{r}}(\{\boldsymbol{f}(\boldsymbol{\theta}^{(1)}), \ldots, \boldsymbol{f}(\boldsymbol{\theta}^{(n)})\})}{\partial f_k(\boldsymbol{\theta}^{(i)})}}_{a_i:1\times 1} \underbrace{\frac{\partial f_k(\boldsymbol{\theta}^{(i)})}{\partial \boldsymbol{\theta}^{(i)}}}_{\boldsymbol{B}:1\times d}. \tag{30}$$

Once $\boldsymbol{d}^{(i)}$ is calculated, solutions $\boldsymbol{\theta}^{(i)}$'s are updated by gradient ascent, $\boldsymbol{\theta}^{(i)} \leftarrow \boldsymbol{\theta}^{(i)} + \eta \boldsymbol{d}^{(i)}$, where $\eta$ is a learning rate. The term $\boldsymbol{B}$ in Equation (30) can be easily estimated via backpropagation. The main challenge lies in estimating the first term $a_i$. As proposed by Emmerich et al. (2007), index sets are categorized based on differentiability: $\mathbb{Z}$ includes subvectors with zero partial gradients, $\mathbb{U}$ contains subvectors with undefined or indeterminate gradients, and $\mathbb{P}$ has subvectors with positive gradients. This classification leads to many zero-gradient terms, improving efficiency. Then, non-zero terms are computed using a fast dimension-sweeping algorithm (Beume et al., 2009; Guerreiro et al., 2012). This method applies to bi-, tri-, and four-objective problems with time complexities of $\Theta(mn + n\log n)$ for $m = 2$ or $3$ and $\Theta(mn + n^2)$ for $m = 4$, where $m$ is the number of objectives and $n$ is the number of solutions. To further enhance convergence, the Newton-hypervolume method (Sosa Hernández et al., 2020) incorporates second-order information for faster updates. Deist et al. (2021) apply the hypervolume gradient methods for deep learning tasks such as medical image classification with more than thousands of parameters.

Inspired by Stein Variational Gradient Descent (SVGD) (Liu & Wang, 2016), MOO-SVGD (Liu et al., 2021d) proposes a different approach to maintain diversity while enhancing convergence. The update direction for the $i$-th solution $\boldsymbol{\theta}^{(i)}$ is given by:

$$\boldsymbol{d}^{(i)} = -\sum_{j=1}^{n} \boldsymbol{g}^*(\boldsymbol{\theta}^{(j)}) \kappa(\boldsymbol{f}(\boldsymbol{\theta}^{(i)}), \boldsymbol{f}(\boldsymbol{\theta}^{(j)})) + \mu \nabla_{\boldsymbol{\theta}^{(i)}} \kappa(\boldsymbol{f}(\boldsymbol{\theta}^{(i)}), \boldsymbol{f}(\boldsymbol{\theta}^{(j)})). \tag{31}$$

where $\boldsymbol{g}^*(\boldsymbol{\theta}^{(j)}) = \sum_{i=1}^{m} \lambda_i^* \nabla f_i(\boldsymbol{\theta}^{(j)})$, and $\{\lambda_i^*\}_{i=1}^{m}$ are the objective weights obtained by MGDA (Sener & Koltun, 2018) (i.e., problem (13)). $\kappa(\cdot, \cdot)$ represents a kernel function, often chosen as the radial basis function (RBF) kernel. The first term in Equation (31) drives the particles toward the target distribution, while the second term acts as a repulsive force that promotes diversity.

While the methods discussed above address scenarios with more solutions than objectives ($n > m$), recent work has begun to explore the converse case, where objectives outnumber solutions ($m > n$) (Lin et al., 2025c; Li et al., 2025; Liu et al., 2024c; Ding et al., 2024). These approaches assign each objective to a solution $\boldsymbol{\theta} \in \{\boldsymbol{\theta}^1, \ldots, \boldsymbol{\theta}^n\}$ that achieves the smallest objective value for that particular objective. The goal is then to minimize an aggregation of these smallest objective values, typically either their sum or their maximum. The Few for Many (F4M) framework (Lin et al., 2025c; Liu et al., 2024c), for example, employs a minimax strategy by minimizing the maximum value, which is formulated as:

$$\min_{\boldsymbol{\theta}^{(1)}, \ldots, \boldsymbol{\theta}^{(n)}} \max_{i\in[m]} \min_{\boldsymbol{\theta}\in\{\boldsymbol{\theta}^{(1)}, \ldots, \boldsymbol{\theta}^{(n)}\}} f_i(\boldsymbol{\theta}). \tag{32}$$

Different from F4M, Sum of Minimum (SoM) (Ding et al., 2024) uses the sum aggregation, formulated as:

$$\min_{\boldsymbol{\theta}^{(1)}, \ldots, \boldsymbol{\theta}^{(n)}} \sum_{i=1}^{m} \left( \min_{\boldsymbol{\theta}\in\{\boldsymbol{\theta}^{(1)}, \ldots, \boldsymbol{\theta}^{(n)}\}} f_i(\boldsymbol{\theta}) \right). \tag{33}$$

MosT (Li et al., 2025) frames the problem as a bilevel optimization task, with the outer loop using MGDA to optimize solutions and the inner loop solving an optimal transport problem to assign objectives to solutions.

### 4.3 Discussions

Preference-vector methods are most suitable when the desired trade-offs are known in advance. They decompose the finite-set problem into preference-specific subproblems, making them easy to parallelize and simple

Table 3: Representative methods for finding a finite set of solutions. "Preference" denotes fixed vectors (✓), no vectors (✗), or adaptive vectors. "Exact" applies only to fixed-vector alignment; "Coupled" means jointly optimized solutions.

| Method | Pref. | Exact | Coupled | Mechanism |
|---|---|---|---|---|
| PMTL (Lin et al., 2019a) | ✓ | ✗ | ✗ | sector constraint |
| EPO (Mahapatra & Rajan, 2020; 2021) | ✓ | ✓ | ✗ | uniformity function |
| WC-MGDA (Momma et al., 2022) | ✓ | ✓ | ✗ | Tchebycheff dual |
| PMGDA (Zhang et al., 2024b) | ✓ | ✓ | ✗ | direction constraint |
| FERERO (Chen et al., 2024a) | ✓ | ✓ | ✗ | constrained MGDA |
| GMOOAR (Chen & Kwok, 2022) | Adapt. | – | ✓ | HV / uniformity (bi-level) |
| UMOD (Zhang et al., 2024a) | Adapt. | – | ✓ | max-min distance |
| GradHV (Wang et al., 2017) | ✗ | – | ✓ | hypervolume gradient |
| MOO-SVGD (Liu et al., 2021d) | ✗ | – | ✓ | SVGD repulsion |
| F4M (Lin et al., 2025c) | ✗ | – | ✓ | minimax ($m > n$) |
| SoM (Ding et al., 2024) | ✗ | – | ✓ | sum-of-min ($m > n$) |
| MosT (Li et al., 2025) | ✗ | – | ✓ | optimal transport ($m > n$) |

to control. PMTL gives sector-level control, while EPO, WC-MGDA, PMGDA, and FERERO aim for stricter alignment with fixed preference vectors. Their main limitation is that uniformly chosen preferences do not necessarily produce uniformly distributed solutions, especially when the Pareto front is non-convex or highly irregular. Adaptive methods such as GMOOAR and UMOD reduce this burden by moving reference vectors or target positions during training.

Preference-free methods are more appropriate when the front shape is unknown or when the goal is to obtain an exploratory set of diverse solutions. By optimizing all candidates jointly, they can allocate solutions according to the geometry of the learned front, but this also couples the candidates and requires maintaining multiple models during training. GradHV provides a principled way to balance convergence and diversity through the hypervolume indicator, while MOO-SVGD explicitly promotes diversity through particle repulsion. When the number of objectives exceeds the desired solution budget ($m > n$), F4M, SoM, and MosT are more targeted choices because they seek a small representative set rather than trying to cover the whole front.

# 5 Finding an Infinite Set of Solutions

While a finite set of solutions can only provide a discrete approximation of the Pareto front, many applications require the ability to obtain solutions corresponding to any user preference, effectively representing the entire Pareto set. Directly learning an infinite number of solutions individually is impractical. Ma et al. (2020) initially investigated deriving an infinite set of solutions by approximating the Pareto set with first-order expansion around discrete Pareto-optimal solutions. However, this method has several drawbacks: (1) The approximation error grows when the solutions are widely spaced, (2) First-order approximations perform poorly in high-dimensional objective spaces, and (3) The approach requires solving a linear programming problem for updates, which can be computationally challenging. To address these issues, many methods have been introduced that leverage neural networks to learn mappings from user preferences to solutions directly, enabling the capture of the entire PS. These methods rely on designing efficient network architectures and implementing effective training strategies.

In Section 5.1, we introduce various network structures designed to capture the entire PS through preference-based mappings. Specifically, current algorithms mainly use three types of network structures: (1) Hypernetwork in Section 5.1.1, (2) Preference-Conditioned Network in Section 5.1.2, and (3) Model Combination in Section 5.1.3. An illustration of these methods is shown in Figure 6. Table 4 compares these structures in terms of parameter overhead, scalability, and expressiveness. Section 5.2 discusses training strategies for these network structures.

Table 4: Network structures for learning an infinite set of solutions. Parameter overhead is measured relative to a single base model; scalability denotes applicability to large backbones.

| Structure | Param. overhead | Scalability | Expressiveness | Representative methods |
|---|---|---|---|---|
| Hypernetwork (§5.1.1) | high | low | high | PHN; CPMTL; PHN-HVI; Hyper-Trans |
| Preference-conditioned (§5.1.2) | low | high | low–medium | COSMOS; YOTO; Raychaudhuri et al. |
| Model combination (§5.1.3) | medium | medium–high | medium–high | PaMaL; LORPMAN; Panacea; PaLoRA |

## 5.1 Network Structure

### 5.1.1 Hypernetwork

Hypernetwork is a neural network that generates the parameters for another target network (Ha et al., 2016). This idea has been leveraged by the Pareto Hypernetwork (PHN) (Navon et al., 2021) and Controllable Pareto Multi-Task Learning (CPMTL) (Lin et al., 2020) to learn the entire Pareto set, where the input is the user preference $\boldsymbol{\alpha}$ and the output is the target network's parameters. The hypernetwork usually consists of several MLP layers, and has been adopted in many subsequent approaches such as PHN-HVI (Hoang et al., 2023) and SUHNPF (Gupta et al., 2022). Recently, Tuan et al. (2024) propose using a transformer architecture as the hypernetwork and report improvements over MLP-based hypernetworks in their Pareto-set learning experiments.

A primary limitation of hypernetwork-based algorithms is their sizes. Since the output dimension matches the number of parameters in the target network, hypernetworks are often much larger than the base networks, limiting their applicability to large models. Some methods address this limitation by employing hypernetworks with chunking (Navon et al., 2021; Lin et al., 2020). Chunking involves partitioning the parameter space into smaller, more manageable segments, enabling the hypernetwork to generate parameters more efficiently and scalability. This approach reduces the overall size of the hypernetwork while preserving its capability to produce accurate parameters for the target network.

### 5.1.2 Preference-Conditioned Network

Instead of using a hypernetwork to generate weights, the original model can be directly modified to incorporate preferences. COSMOS (Ruchte & Grabocka, 2021) suggests adding preferences as an additional input to the model by combining user preference $\boldsymbol{\alpha}$ with input data $\boldsymbol{x}$, thereby increasing the input dimension of the original model.

However, such input-based conditioning has limited ability to produce diverse solutions. Some studies (Dosovitskiy & Djolonga, 2020; Chen & Kwok, 2022; Raychaudhuri et al., 2022) employ the Feature-wise Linear Modulation (FiLM) layer (Perez et al., 2018) to condition the network. The FiLM layer works by applying an affine transformation to the feature maps. Specifically, given a feature map $\boldsymbol{u}$ with $C$ channels, we use an MLP to generate conditioning parameters $\boldsymbol{\gamma} \in \mathbb{R}^C$ and $\boldsymbol{\beta} \in \mathbb{R}^C$ based on the given preference $\boldsymbol{\alpha}$. Then, the FiLM layer modifies the features as $\boldsymbol{u}'_c = \gamma_c \cdot \boldsymbol{u}_c + \beta_c$, where $\boldsymbol{u}_c$ is the $c$-th channel of $\boldsymbol{u}$. This conditioning mechanism lets the preference vector modulate intermediate feature channels rather than only augmenting the input layer. Raychaudhuri et al. (2022) also propose leveraging another network to predict the network architecture. It dynamically adapts the model's structure according to user preferences, allowing for more flexible and efficient learning.

### 5.1.3 Model Combination

Methods based on model combination construct a composite model by integrating multiple individual models, thereby offering an effective way to introduce diversity. PaMaL (Dimitriadis et al., 2023) achieves this by learning several base models and combining them through a weighted sum of their parameters based on user

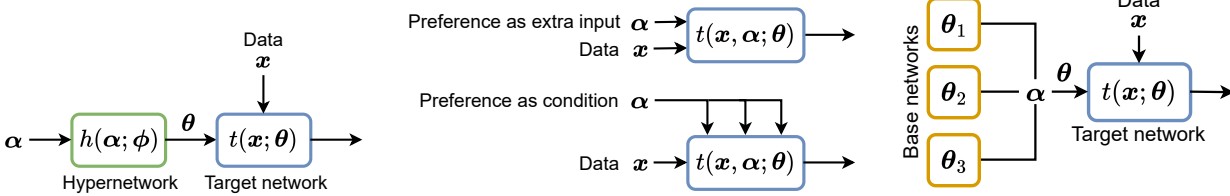

(a) Methods based on the hyper-network.

(b) Methods based on preference-conditioned network.

(c) Methods based on model combination.

Figure 6: Illustration of different structures to learn an infinite number of solutions.

preferences:

$$\boldsymbol{\theta}(\boldsymbol{\alpha}) = \sum_{i=1}^{m} \alpha_i \boldsymbol{\theta}_i. \tag{34}$$

Although this incorporates more parameters than methods based on preference-conditioned networks, PaMaL enhances diversity and provides users with a broader range of choices.

Despite its benefits, learning multiple models simultaneously may become inefficient when handling a large number of objectives. To mitigate this issue, LORPMAN (Chen & Kwok, 2024a) proposes learning a full-rank model $\boldsymbol{\theta}_0 \in \mathbb{R}^{p \times q}$ and $m$ low-rank models $\boldsymbol{B}_i \boldsymbol{A}_i$'s, where $\boldsymbol{B}_i \in \mathbb{R}^{p \times r}$ and $\boldsymbol{A}_i \in \mathbb{R}^{r \times q}$. Given user preference $\boldsymbol{\alpha}$, the composite model is given by:

$$\boldsymbol{\theta}(\boldsymbol{\alpha}) = \boldsymbol{\theta}_0 + s \sum_{i=1}^{m} \alpha_i \boldsymbol{B}_i \boldsymbol{A}_i, \tag{35}$$

where $s$ is a scaling factor that adjusts the impact of the low-rank components. This construction is related to LoRA-style adaptation (Hu et al., 2021), but its role is different from composing existing LoRA modules. LoRAHub (Huang et al., 2023) combines already-trained LoRA modules for cross-task generalization without additional model parameters or gradients, while Mixture of LoRA Experts (MoLE) (Wu et al., 2024) treats trained LoRAs as experts and learns gating weights for flexible composition. In contrast, LORPMAN jointly learns a shared full-rank model and preference-weighted low-rank components as a continuous Pareto-manifold subspace, so different preferences correspond to different points in that learned subspace rather than to routed combinations of pre-trained adapters. A similar approach is also mentioned in (Dimitriadis et al., 2025). To further enhance parameter efficiency, Chen & Kwok (2024b) introduce a preference-aware tensor multiplication:

$$\boldsymbol{\theta}(\boldsymbol{\alpha}) = \boldsymbol{\theta}_0 + s\boldsymbol{C} \times_1 \boldsymbol{A} \times_2 \boldsymbol{B} \times_3 \boldsymbol{\alpha}, \tag{36}$$

where $\boldsymbol{C} \in \mathbb{R}^{r \times r \times m}$ is a learnable core tensor, $\boldsymbol{A} \in \mathbb{R}^{r \times p}$, $\boldsymbol{B} \in \mathbb{R}^{r \times q}$ are learnable matrices, and $\times_u$ denotes mode-$u$ tensor multiplication (Kolda & Bader, 2009). For large-scale models such as LLMs, Zhong et al. (2024) propose Panacea, which uses SVD-LoRA, a preference-conditioned low-rank parameterization in which preferences are injected through singular values:

$$\boldsymbol{\theta}(\boldsymbol{\alpha}) = \boldsymbol{\theta}_0 + \boldsymbol{U}\boldsymbol{\Sigma}\boldsymbol{V}, \tag{37}$$

where $\boldsymbol{\Sigma}$ is a diagonal matrix defined as $\mathrm{diag}(\sigma_1, \ldots, \sigma_r, s\alpha_1, \ldots, s\alpha_m)$, $\{\sigma_i\}_{i=1}^{r}$ and $s$ are learnable scalars. $\boldsymbol{U} \in \mathbb{R}^{p \times (r+m)}$ and $\boldsymbol{V} \in \mathbb{R}^{(r+m) \times q}$ are learnable matrices.

Instead of training new models, Tang et al. (2024) propose to utilize a mixture of experts (Cai et al., 2024) to combine pre-existing models in order to achieve the Pareto set. Given $n$ models, $\boldsymbol{\theta}_1, \ldots, \boldsymbol{\theta}_n$, each trained on different datasets, they are first integrated into a unified base model $\boldsymbol{\theta}_0$. For some modules in the network, differences between the individual models and the base model (i.e., $\boldsymbol{\theta}_i - \boldsymbol{\theta}_0$'s) are maintained. These differences are treated as experts within the framework. Then, a mixture of experts is applied, where a gating network conditioned on user preferences determines the weighting of these experts. It enables the generation of diverse models by applying different weighted combinations of experts to align with specific user preferences.

## 5.2 Training Strategy

This section outlines the training approach for the parameters (denoted $\phi$) of the network structures introduced in Section 5.1. Specifically, $\phi$ refers to the parameters of the hypernetwork in Section 5.1.1, parameters of the preference-conditioned network in Section 5.1.2, or parameters of the base models or low-rank models in Section 5.1.3. Current algorithms follow a similar approach: sample user preferences and optimize the structure parameters to generate networks that align with those preferences. The training objective can be written as:

$$\min_{\phi} \mathbb{E}_{\boldsymbol{\alpha} \sim \Delta_{m-1}} \mathbb{E}_{(\boldsymbol{x}, \boldsymbol{y}) \sim \mathcal{D}} \left[ \tilde{g}_{\boldsymbol{\alpha}}(\boldsymbol{\ell}(t(\boldsymbol{x}; \boldsymbol{\theta}(\boldsymbol{\alpha}; \phi)), \boldsymbol{y})) \right], \tag{38}$$

where $(\boldsymbol{x}, \boldsymbol{y})$'s are training data sampled from dataset $\mathcal{D}$, $\boldsymbol{\ell}(\cdot, \cdot)$ is the multi-objective loss function and $\tilde{g}_{\boldsymbol{\alpha}}(\cdot)$ denotes a preference-dependent scalar training criterion induced by a scalarization or MOO solver. For gradient-based training, it is implemented as a differentiable criterion or a differentiable/smoothed surrogate; when the original criterion is nonsmooth, such as the standard Tchebycheff scalarization, methods can use subgradients or smooth approximations rather than requiring the exact $\tilde{g}_{\boldsymbol{\alpha}}$ to be differentiable everywhere. In general, most optimization algorithms discussed in Sections 2.2 and 4 can be adopted. Below, we summarize the algorithms adopted by the existing methods.

- Scalarization: As outlined in Section 2.2, scalarization combines multiple objectives into one scalar objective. Linear scalarization is common in most algorithms, such as PHN (Navon et al., 2021), COSMOS (Ruchte & Grabocka, 2021), PAMAL (Dimitriadis et al., 2023), and LORPMAN (Chen & Kwok, 2024a). Alternatives like Tchebycheff and smooth Tchebycheff scalarization (Lin et al., 2024b) can also be adopted.

- Preference-Aware Weighting Methods: As discussed in Section 4.1, these methods incorporate user preferences into optimization. PHN (Navon et al., 2021) employs the Exact Pareto Optimal (EPO) solver (Mahapatra & Rajan, 2020) to find Pareto-optimal solutions aligned with preferences. CPMTL (Lin et al., 2020) uses a constrained approach inspired by PMTL (Lin et al., 2019a).

- Hypervolume Maximization: As discussed in Section 4.2, hypervolume maximization can optimize both diversity and convergence of a solution set. PHN-HVI (Hoang et al., 2023) leverages hypervolume maximization to optimize the structural parameters $\phi$. In each iteration, it samples a set of preference vectors and generates solutions based on these vectors. Subsequently, hypervolume maximization is applied to promote both the diversity and convergence of the solutions.

## 5.3 Discussions

Table 4 highlights the main expressiveness–scalability trade-off. Hypernetworks are flexible because they generate preference-specific target models, but their output dimension scales with the target network, making them difficult to use with very large backbones. Preference-conditioned networks add little overhead and are therefore the most scalable, but their restricted parameterization can limit Pareto-set diversity.

Model-combination methods sit between these extremes. PaMaL (Dimitriadis et al., 2023) maintains one base model per objective, so its overhead grows with $m$; low-rank variants such as LORPMAN (Chen & Kwok, 2024a), Panacea (Zhong et al., 2024), and PaLoRA (Dimitriadis et al., 2025) reduce this overhead and are the strongest candidates for large models. For training, linear scalarization is the simplest default, preference-aware weighting is better when alignment to user preferences matters, and HV-based criteria are useful when global set quality is the main goal.

# 6 Theories

## 6.1 Convergence of Gradient-Balancing Methods in Section 3.2

This section reviews the convergence analysis for gradient-balancing methods in Section 3.2. This is first examined under the deterministic gradient setting (i.e., full-batch gradient). Subsequently, it is analyzed

Table 5: Convergence of stochastic gradient MOO algorithms. LS/GS denote $L$-smooth/generalized $L$-smooth objectives; BG/BF denote bounded gradient/function-value assumptions. Complexity is the sample complexity for an $\epsilon$-accurate Pareto stationary point; CA denotes conflict-avoidant distance.

| Method | Batch Size | Assumption | Complexity | Bounded CA |
|---|---|---|---|---|
| SMG (Liu & Vicente, 2021) | $\mathcal{O}(\epsilon^{-2})$ | LS, BG | $\mathcal{O}(\epsilon^{-4})$ | ✗ |
| CR-MOGM (Zhou et al., 2022) | $\mathcal{O}(1)$ | LS, BF, BG | $\mathcal{O}(\epsilon^{-2})$ | ✗ |
| MoCo (Fernando et al., 2023) | $\mathcal{O}(1)$ | LS, BF, BG | $\mathcal{O}(\epsilon^{-2})$ | ✗ |
| PSMGD (Xu et al., 2025a) | $\mathcal{O}(1)$ | LS, BF, BG | $\mathcal{O}(\epsilon^{-2})$ | ✗ |
| MoDo (Chen et al., 2023c) | $\mathcal{O}(1)$ | LS, BG | $\mathcal{O}(\epsilon^{-2})$ | ✓ |
| SDMGrad (Xiao et al., 2023) | $\mathcal{O}(1)$ | LS, BG | $\mathcal{O}(\epsilon^{-2})$ | ✓ |
| SGSMGrad (Zhang et al., 2025) | $\mathcal{O}(1)$ | GS | $\mathcal{O}(\epsilon^{-2})$ | ✓ |

under the stochastic gradient setting. Note that in non-convex MOO, convergence refers to reaching Pareto stationary (Definition 5).

### 6.1.1 Deterministic Gradient

With the use of deterministic gradients, it has been shown that under mild conditions, MGDA (Sener & Koltun, 2018) can converge to a Pareto-stationary point at a convergence rate of $\mathcal{O}(K^{-1/2})$ (Fliege et al., 2019), where $K$ is the number of iterations. This rate is comparable to that of single-objective optimization (Nesterov, 2013). The fundamental idea is to assess the reduction in each individual objective function. When the current solution is not Pareto-stationary, it can be shown that the common descent direction $\boldsymbol{d}$ identified by MGDA ensures an improvement of at least $\mathcal{O}(\|\boldsymbol{d}\|^2)$ in each objective, which mirrors the situation in single-objective optimization (Fliege et al., 2019). Therefore, the method used in single-objective optimization can be applied to prove the convergence. Other algorithms such as CAGrad (Liu et al., 2021a) and Nash-MTL (Navon et al., 2022) also provide deterministic convergence analyses. PCGrad (Yu et al., 2020) includes a limited theoretical analysis under specific two-task and convex-objective assumptions.

### 6.1.2 Stochastic Gradient

Liu & Vicente (2021) are the first to analyze the convergence of gradient-based MOO algorithms with stochastic gradients. They propose a stochastic version of MGDA (Sener & Koltun, 2018) that computes a common descent direction using stochastic gradients for all objectives and prove convergence to Pareto-optimal solutions under the assumption of convex objective functions. However, due to the inherent bias of common descent direction $\boldsymbol{d}$ introduced by stochastic gradient estimations, their analysis requires the use of an increasing batch size that grows linearly with the number of iterations to ensure convergence. This requirement can be impractical in real-world applications, as large batch sizes lead to increased computational costs and memory usage.

To overcome this limitation, Zhou et al. (2022) propose Correlation-Reduced Stochastic Multi-Objective Gradient Manipulation (CR-MOGM). It addresses the bias in the common descent direction by introducing a smoothing technique on weight coefficient $\boldsymbol{\lambda}$. In CR-MOGM, $\boldsymbol{\lambda}^{(k)}$ at iteration $k$ is updated using a moving average:

$$\boldsymbol{\lambda}^{(k)} = (1-\gamma)\hat{\boldsymbol{\lambda}}^{(k)} + \gamma\boldsymbol{\lambda}^{(k-1)}, \tag{39}$$

where $\gamma \in [0, 1]$ is a smoothing factor, $\hat{\boldsymbol{\lambda}}^{(k)}$ is the weight vector obtained by the MOO solver (such as MGDA) at iteration $k$ using the current stochastic gradients, and $\boldsymbol{\lambda}^{(k-1)}$ is the smoothed weight from the previous iteration. Smoothing reduces the variance in the weight, leading to a more stable and reliable common descent direction. Xu et al. (2025a) further analyze the convergence by updating the objective weights every $\tau$ iterations instead of at each iteration, and achieve the same convergence rate $\mathcal{O}(\epsilon^{-2})$ as CR-MOGM. They also match the convergence rate in stochastic single-objective optimization (Ghadimi & Lan, 2013).

Another approach to address gradient bias is MoCo (Fernando et al., 2023), which introduces a tracking variable $\hat{\boldsymbol{g}}_i^{(k)}$ that approximates the true gradient:

$$\hat{\boldsymbol{g}}_i^{(k+1)} = \prod_{L_i} \left( \hat{\boldsymbol{g}}_i^{(k)} - \gamma \left( \hat{\boldsymbol{g}}_i^{(k)} - \boldsymbol{g}_i^{(k)} \right) \right), \tag{40}$$

where $\prod_{L_i}$ is the projection to the set $\{\boldsymbol{g} \in \mathbb{R}^d \mid \|\boldsymbol{g}\| \leq L_i\}$, and $L_i$ is the Lipschitz constant of $f_i(\boldsymbol{\theta})$. However, the analysis requires the number of iterations $K$ to be sufficiently large (in the order of $\mathcal{O}(m^{10})$). This high dependency on $m$ makes MoCo less practical for problems involving many objectives. Additionally, the convergence analysis of CR-MOGM and MoCo relies on the assumption that the objective functions have bounded values.

To address these limitations, Chen et al. (2023c) propose the Multi-objective gradient with double sampling algorithm (MoDo). MoDo mitigates bias in stochastic gradient-based MOO methods without the bounded function value assumption, while still relying on $L$-smoothness and bounded stochastic gradients as summarized in Table 5. It introduces a double sampling technique to obtain unbiased estimates of the gradient products needed for updating the weight coefficients. Specifically, at each iteration, MoDo updates the weight vector $\boldsymbol{\lambda}^{(k)}$ using gradients computed on two independent mini-batches:

$$\boldsymbol{\lambda}^{(k+1)} = \prod_{\Delta_{m-1}} \left( \boldsymbol{\lambda}^{(k)} - \eta \boldsymbol{G}^{(k)}(\boldsymbol{z}_1^{(k)})^\top \boldsymbol{G}^{(k)}(\boldsymbol{z}_2^{(k)}) \boldsymbol{\lambda}^{(k)} \right), \tag{41}$$

where $\eta$ is the step size, $\prod_{\Delta_{m-1}}$ is the projection onto the simplex $\Delta_{m-1}$, and $\boldsymbol{G}^{(k)}(\boldsymbol{z}_1^{(k)})$, $\boldsymbol{G}^{(k)}(\boldsymbol{z}_2^{(k)})$ are stochastic gradients evaluated on two independent mini-batches $\boldsymbol{z}_1^{(k)}$ and $\boldsymbol{z}_2^{(k)}$. By reducing the bias in estimating the common descent direction, MoDo achieves convergence guarantees without requiring bounded objective values. Additionally, it guarantees a bounded conflict-avoidant (CA) distance, which is the distance between the estimated update direction and the CA direction defined in problem (12).

Similarly, Xiao et al. (2023) propose the Stochastic Direction-oriented Multi-objective Gradient descent (SDMGrad) algorithm, which introduces a new direction-oriented multi-objective formulation by regularizing the common descent direction within a neighborhood of a target direction (such as the average gradient of all objectives). In SDMGrad, the weight vector $\boldsymbol{\lambda}^{(k)}$ is updated as:

$$\boldsymbol{\lambda}^{(k+1)} = \prod_{\Delta_{m-1}} \left( \boldsymbol{\lambda}^{(k)} - \eta \left[ \boldsymbol{G}^{(k)}(\boldsymbol{z}_1^{(k)})^\top \left( \boldsymbol{G}^{(k)}(\boldsymbol{z}_2^{(k)}) \boldsymbol{\lambda}^{(k)} + \gamma \boldsymbol{g}_0(\boldsymbol{z}_2^{(k)}) \right) \right] \right), \tag{42}$$

where $\eta$ is the step size, $\gamma$ is a regularization factor controlling the proximity to the target direction $\boldsymbol{g}_0(\boldsymbol{z}_2^{(k)})$. By incorporating this regularization, SDMGrad effectively balances optimization across objectives while guiding the overall descent direction. Zhang et al. (2025) further consider the convergence under generalized $L$-smoothness (Li et al., 2023) and without bounded gradient assumption. A summary of the convergence results of existing stochastic gradient MOO algorithms is provided in Table 5.

## 6.2 Generalization

The generalization aspect of multi-objective deep learning remains relatively underexplored compared to its convergence properties. Cortes et al. (2020) analyze the generalization behavior of a specific scalarization approach that minimizes over convex combinations. Súkeník & Lampert (2024) consider a broader class of scalarization methods. They show that the generalization bounds for individual objectives extend to MOO with scalarization. Both studies rely on the Rademacher complexity of the hypothesis class to establish algorithm-independent generalization bounds, which are unaffected by the training process.

Another line of research investigates Tchebycheff scalarization, which focuses on the sample complexity required to achieve a generalization error within $\epsilon$ of the optimal objective value. Formally, given a set of $m$ distributions $\{\mathcal{D}_i\}_{i=1}^m$ and a hypothesis class $\mathcal{H}$, the goal is to find a (possibly randomized) hypothesis $h$ such that

$$\max_{i \in [m]} \ell_{\mathcal{D}_i}(h) \leq \min_{h^* \in \mathcal{H}} \max_{i \in [m]} \ell_{\mathcal{D}_i}(h^*) + \epsilon, \tag{43}$$

where $\ell_{\mathcal{D}_i}(h)$ is the loss of hypothesis $h$ on $\mathcal{D}_i$, $h^*$ is the optimal hypothesis and $\epsilon$ is the error tolerance. It is first formulated by Awasthi et al. (2023) as an open problem in 2023 and has since been addressed by several works (Peng, 2024; Zhang et al., 2024d). Haghtalab et al. (2022) are the first to show the sample complexity lower bound of $\widetilde{\Omega}\left(\frac{v+m}{\epsilon^2}\right)$, where $v = \text{VCdim}(\mathcal{H})$ is the VC-dimension of the hypothesis class $\mathcal{H}$. By using a boosting framework, Peng (2024) gives an algorithm that achieves a sample complexity upper bound of $\widetilde{O}\left(\frac{v+m}{\epsilon^2} \cdot \left(\frac{m}{\epsilon}\right)^{o(1)}\right)$, which is nearly optimal. In a concurrent work, Zhang et al. (2024d) provide a variant of the hedge algorithm under an Empirical Risk Minimization (ERM) oracle access that achieves the optimal sample complexity bound of $\widetilde{O}\left(\frac{v+m}{\epsilon^2}\right)$.

In the online setting, Liu et al. (2024b) provide an adaptive online mirror descent algorithm that achieves $\mathcal{O}\left(\frac{mv}{\sqrt{K}}\right)$ regret, where $K$ is the number of iterations. Using a plain online-to-batch conversion scheme, this algorithm leads to an $\mathcal{O}\left(\frac{m^2 v^2}{\epsilon^2}\right)$ sample complexity, which still lags behind the optimal offline sample complexity of $\widetilde{O}\left(\frac{m+v}{\epsilon^2}\right)$ (Zhang et al., 2024d). It is interesting to see whether it is possible to use an online learner with proper online-to-batch conversion schemes that is able to match the optimal offline sample complexity.

MoDo (Chen et al., 2023c) introduces a different perspective by leveraging algorithm stability to derive algorithm-dependent generalization error bounds of gradient balancing algorithms.

# 7 Applications

In this section, we review key application domains where gradient-based MOO has been successfully applied, spanning computer vision, neural architecture search, recommendation systems, large language models, and other emerging areas. For each domain, we connect representative works to the trade-offs and method families discussed earlier, so that the section functions as a compact bridge from the taxonomy to practice.

## 7.1 Computer Vision

The most representative application of MOO in computer vision is multi-task dense prediction (Vandenhende et al., 2021), which aims to train a model for simultaneously dealing with multiple dense prediction tasks (such as semantic segmentation, monocular depth estimation, and surface normal estimation) and has been successfully applied in autonomous driving (Ishihara et al., 2021; Liang et al., 2023). Since the encoder in computer vision usually contains large number of parameters, sharing the encoder across different tasks can significantly reduce the computational cost but may cause conflicts among tasks, leading to a performance drop in some of the tasks. Hence, many methods are proposed to address the issue from the perspective of MOO, which use scalarization to balance multiple losses (Xu et al., 2022; Ye & Xu, 2022; Lin et al., 2024a; 2025a; Li et al., 2024a). This setting illustrates a common use of single-solution MOO methods: balancing dense prediction objectives within a shared encoder. A comprehensive survey on multi-task dense prediction is in (Vandenhende et al., 2021). Besides the dense prediction task, MOO is also useful in other computer vision tasks such as point cloud (Xie et al., 2023; 2024; Wang et al., 2024b), medical image denoising (Kyung et al., 2024), and pose estimation (Ye et al., 2024a).

## 7.2 Neural Architecture Search

Neural Architecture Search (NAS), which aims to design the architecture of neural networks automatically, has gained significant interest recently (Ren et al., 2021). Due to the complex application scenarios in the real world, recent works consider multiple objectives beyond just accuracy. For example, to search an efficient architecture for deployment on platforms with limited resources, many studies incorporate resource-constraint objectives such as the parameter size, FLOPs, and latency. These works are mainly based on gradient-based NAS methods such as DARTS (Liu et al., 2019a) and formulate it as a multi-objective optimization problem. Among them, Wu et al. (2019); Cai et al. (2019); Wu et al. (2021); Wang et al. (2021a); Yue et al. (2022) employ scalarization to identify a single solution, where the task and efficiency objectives are combined with fixed weights. However, altering the weights of the objectives needs a complete

rerun of the search, which is computationally intensive. Therefore, Sukthanker et al. (2025) propose to learn a mapping from a preference vector to an architecture using a hypernetwork, enabling to provide the entire PS without the need for a new search. This setting illustrates how MOO is used to trade prediction quality against deployment cost, with both scalarized single-solution search and preference-conditioned Pareto-set learning appearing in recent work.

### 7.3 Recommendation Systems

Beyond just focusing on accuracy, recommendation systems often incorporate additional quality metrics such as novelty, diversity, serendipity, popularity, and others (Zheng & Wang, 2022). These factors naturally frame the problem as a MOO problem. Traditionally, many methods have employed scalarization techniques to assign weights to the different objectives (Di Noia et al., 2017; Patil et al., 2021; Lacerda, 2017). In recent developments, gradient-balancing approaches are applied to address multi-objective optimization within recommendation systems to learn a single recommendation model (Milojkovic et al., 2019; Lin et al., 2019b; Mitrevski et al., 2020; Yang et al., 2024a). Additionally, some research efforts aim to identify finite Pareto sets (Xie et al., 2021) and infinite Pareto sets (Ge et al., 2022; Wilm et al., 2024), enabling the provision of personalized recommender models that cater to varying user preferences. This domain illustrates both single-solution balancing and Pareto-set learning, because different users or products may prefer different trade-offs among accuracy, diversity, novelty, and related metrics.

### 7.4 Large Language Models (LLMs)

Recently, there has been a growing trend of incorporating multi-objective optimization into the training of large language models (LLMs), including multi-task fine-tuning and multi-objective alignment.

Due to the powerful transferability of LLMs, users can fine-tune LLMs to specific downstream tasks or scenarios. This approach separates fine-tuning on each task, causing extensive costs in training and difficulties in deployment. MFTCoder (Liu et al., 2024a) proposes a multi-task fine-tuning framework that enhances the coding capabilities of LLMs by addressing data imbalance, varying difficulty levels, and inconsistent convergence speeds across tasks. CoBa (Gong et al., 2024) studies multi-task fine-tuning for LLMs and balances task convergence with minimal computational overhead.

Aligning with multi-dimensional human preferences (such as helpfulness, harmfulness, humor, and conciseness) is essential for customizing responses to users' needs, as users typically have diverse preferences for different aspects. MORLHF (Wu et al., 2023) and MODPO (Zhou et al., 2024a) train an LLM for every preference configuration by linearly combining multiple (implicit) reward models. To avoid retraining, some methods train multiple LLMs separately for each preference dimension and merge their parameters (Rame et al., 2023; Jang et al., 2023) or output logits (Shi et al., 2024) to deal with different preference requirements. However, these methods need to train and store multiple LLMs, leading to huge computational and storage costs. To improve efficiency, several preference-aware methods are proposed to fine-tune a single LLM for varying preferences by incorporating the relevant coefficients into the input prompts (Wang et al., 2024a; Guo et al., 2024b; Yang et al., 2024b) or model parameters (Zhong et al., 2024; Lin et al., 2025b). This line of work connects MOO to multi-dimensional preference alignment: because training and storing a separate LLM for each preference is costly, recent methods emphasize preference-conditioned or parameter-efficient ways to cover different trade-offs.

### 7.5 Miscellaneous

In addition to the applications mentioned above, multi-objective optimization has been applied to a variety of other deep learning scenarios, including meta-learning (Yu et al., 2023; Wang et al., 2021b), federated learning (Hu et al., 2022; Askin et al., 2024; Kang et al., 2024), long-tailed learning (Li et al., 2024b; Zhao et al., 2024; Zhou et al., 2024b), continual learning (Lai et al., 2025), diffusion models (Hang et al., 2023; Go et al., 2023; Xu et al., 2025b; Yao et al., 2024), neural combinatorial optimization (Li et al., 2020; Lin et al., 2022b; Wang et al., 2023b; Chen et al., 2023b;a), GFlowNets (Jain et al., 2023; Zhu et al., 2023), multi-agent systems (Rădulescu et al., 2020), LLM pruning (Chen et al., 2025) and physics informed neural

Table 6: Benchmark datasets in multi-objective deep learning.

| Dataset | Description | #Obj. | #Samples |
|---|---|---|---|
| NYUv2 (Silberman et al., 2012) | indoor scene understanding | 3 | $1,449$ |
| Taskonomy (Zamir et al., 2018) | indoor scene understanding | 26 | $\approx 4M$ |
| Cityscapes (Cordts et al., 2016) | urban scene understanding | 2 | $3,475$ |
| QM9 (Ramakrishnan et al., 2014) | molecular property prediction | 11 | 130K |
| Office-31 (Saenko et al., 2010) | image classification | 3 | $4,110$ |
| Office-Home (Venkateswara et al., 2017) | image classification | 4 | $15,500$ |
| CelebA (Liu et al., 2015) | image classification | 40 | $\approx 202K$ |
| CIFAR-100 (Krizhevsky & Hinton, 2009) | image classification | 20 | 60K |
| XTREME (Hu et al., 2020) | multilingual learning | 9 | $\approx 597K$ |
| PKU-SafeRLHF (Ji et al., 2024) | two-dimensional preference data | 2 | $\approx 82K$ |
| UltraFeedBack (Cui et al., 2023) | multi-dimensional preference data | 4 | $\approx 64K$ |
| HelpSteer2 (Wang et al., 2024c) | multi-attributes preference data | 5 | $\approx 21K$ |

networks (PINNs) (Hwang & Lim, 2024; Liu et al., 2025). These applications highlight the versatility of MOO in addressing diverse challenges within deep learning.

## 8 Resources

In this section, we introduce some benchmark datasets and open-source libraries for gradient-based MOO in deep learning. The datasets below are selected to cover several objective structures used throughout the survey, from dense prediction and many-attribute classification to multi-dimensional LLM preference data. The libraries provide practical implementations for the two main usage modes emphasized in our taxonomy: training a single balanced model and exploring finite or continuous Pareto sets.

### 8.1 Datasets

Below, we provide details on the benchmark datasets, which are summarized in Table 6:

- **NYUv2** dataset (Silberman et al., 2012) is for indoor scene understanding. It has three tasks (i.e., semantic segmentation, depth estimation, and surface normal prediction) with 795 training and 654 testing samples.

- **Taskonomy** dataset (Zamir et al., 2018) is obtained from 3D scans of about 600 buildings and contains 26 tasks (such as semantic segmentation, depth estimation, surface normal prediction, keypoint detection, and edge detection) with about 4 million samples.

- **Cityscapes** dataset (Cordts et al., 2016) is for urban scene understanding. It has two tasks (i.e., semantic segmentation and depth estimation) with $2,975$ training and 500 testing samples.

- **QM9** dataset (Ramakrishnan et al., 2014) is for molecular property prediction with 11 tasks. Each task performs regression on one property. It contains $130,000$ samples.

- **Office-31** dataset (Saenko et al., 2010) contains $4,110$ images from three domains (tasks): Amazon, DSLR, and Webcam. Each task has 31 classes.

- **Office-Home** dataset (Venkateswara et al., 2017) contains $15,500$ images from four domains (tasks): artistic images, clipart, product images, and real-world images. Each task has 65 object categories collected under office and home settings.

- **CelebA** (Liu et al., 2015) is a large-scale face attribute dataset comprising $202,599$ face images. Each image is annotated with 40 binary attributes, resulting in 40 distinct binary classification tasks.

- **CIFAR-100** (Krizhevsky & Hinton, 2009) is an image classification dataset with 100 classes, containing 50K training and 10K testing images. In multi-task learning benchmarks, CIFAR-100 is often

partitioned into 20 tasks according to its 20 coarse categories, each task being a 5-class classification problem.

- **XTREME** benchmark (Hu et al., 2020) aims to evaluate the cross-lingual generalization abilities of multilingual representations. It contains 9 tasks in 40 languages, including 2 classification tasks, 2 structure prediction tasks, 3 question-answering tasks, and 2 sentence retrieval tasks.

- **PKU-SafeRLHF** dataset (Ji et al., 2024) focuses on safety alignment in LLMs, containing $82,118$ question-answering pairs with the annotations of helpfulness and harmlessness.

- **UltraFeedBack** dataset (Cui et al., 2023) contains $63,967$ prompts, each corresponding to four responses from different LLMs. Each response has 4 aspects of annotations, namely instruction-following, truthfulness, honesty, and helpfulness, generated by GPT-4.

- **HelpSteer2** dataset (Wang et al., 2024c) contains $21,362$ samples. Each sample contains a prompt and a response with 5 human-annotated attributes (i.e., helpfulness, correctness, coherence, complexity, and verbosity), each ranging between 0 and 4 where higher means better for each attribute.

### 8.2 Libraries

Two popular libraries, LibMTL[1] (Lin & Zhang, 2023) and LibMOON[2] (Zhang et al., 2024c), provide unified environments for implementing and fairly evaluating MOO methods. Both are built on PyTorch (Paszke et al., 2019) and feature modular designs, enabling flexible development of new methods or application of existing ones to new scenarios. LibMTL (Lin & Zhang, 2023) mainly focuses on finding a single Pareto-optimal solution and supports 24 methods introduced in Section 3 and 6 benchmark datasets. LibMOON (Zhang et al., 2024c) mainly focuses on exploring the whole Pareto set. It includes over 20 methods for obtaining a finite set of solutions (introduced in Section 4) or infinite set of solutions (introduced in Section 5).

## 9 Challenges and Future Directions

Despite significant progress in applying MOO in deep learning, both in learning a single model and in learning a Pareto set of models, several challenges remain.

**Theoretical Understanding.** While practical methods for MOO in deep learning have seen significant progress, their theoretical foundations remain relatively underexplored. Most existing research focuses on analyzing the convergence of MOO algorithms to stationary points, while generalization error, which is critical for evaluating performance on unseen data, has received less attention. For instance, Chen et al. (2023c), as discussed in Section 6.2, provide the algorithm-dependent generalization analysis. Extending this to a broader range of algorithms could offer deeper insights into how different techniques affect generalization. For Pareto set learning algorithms, there is limited theoretical understanding of how effectively existing algorithms can approximate Pareto sets. Zhang et al. (2023b) established a generalization bound for HV-based Pareto set learning. However, the effect of various network architectures (Section 5.1) on approximating Pareto sets remain unclear. Additionally, it is uncertain how effective current training strategies (Section 5.2) are in approximating the Pareto set.

**Reducing Gradient Balancing Costs.** While gradient balancing methods in Section 3.2 are widely used to mitigate objective conflict, they come with significant computational overhead. Despite the introduction of some practical speedup strategies (discussed in Section 3.2.3), existing approaches remain insufficiently efficient. A deeper understanding of the optimization differences between gradient balancing, linear scalarization, and loss balancing is crucial. This insight could facilitate the integration of gradient balancing with linear scalarization and loss balancing, reducing computational overhead significantly and enabling its application in large-scale training scenarios.

---

[1] https://github.com/median-research-group/LibMTL
[2] https://github.com/xzhang2523/libmoon

**Dealing with Large Number of Objectives.** Some real-world problems involve handling a large number of objectives, which poses significant challenges for current Pareto set learning algorithms in Section 5. As the number of objectives grows, the preference vector space expands exponentially, making it challenging for existing random sampling-based techniques to effectively learn the mapping between preference vectors and solutions. Replacing random sampling with systematic preference exploration, such as methods used in MORL (Röpke et al., 2025), can enable more efficient coverage of the solution set. Additionally, exploring methods to reduce or merge objectives based on their properties can be a promising approach to minimize the total number of objectives. Finally, adopting a utility-based approach (Roijers et al., 2013; Hayes et al., 2022) is promising; by utilizing priors on user utility, learning can be restricted to the significantly smaller subset.

**Distributed Training.** Most existing gradient-based MOO studies report experiments on single-GPU or single-machine settings, but this should not be interpreted as an inherent incompatibility with distributed training. Standard data/model parallelism can often be applied at the autograd level. The main open issues are more specific: gradient-balancing methods require collecting or communicating per-objective gradients efficiently; finite- and infinite-set methods may need to synchronize multiple models or preference-conditioned branches; and bilevel or second-order methods can be difficult to scale because hypergradients, double backpropagation, and distributed Hessian-vector products are less mature in current systems. Additionally, when data for different objectives is distributed across devices and cannot be shared (e.g., due to privacy constraints), collaborative computation of solutions or Pareto sets without direct data sharing remains an important challenge.

**Advancements in Large Language Models (LLMs).** As discussed in Section 7.4, current research in MOO for LLMs largely focuses on the RLHF stage. Expanding MOO techniques to other stages in the LLM lifecycle, such as pre-training and inference, is a valuable direction for future research. Addressing these challenges can lead to models that are better aligned with user needs across their entire development processes. Additionally, user preferences are currently modeled as a preference vector. However, this representation may oversimplify more complex user preferences on LLMs. Exploring advanced methods to capture and represent these intricate preferences can enhance the effectiveness and personalization of LLMs.

**Application in More Deep Learning Scenarios.** While MOO has already been utilized in various deep learning scenarios, as highlighted in Section 7, there remain numerous unexplored areas within this field. Multi-objective characteristics are inherently present in most deep learning problems, as models are typically developed or evaluated based on multiple criteria. Consequently, trade-offs often arise naturally. Leveraging MOO methods to effectively address these trade-offs presents an opportunity for further exploration.

## 10   Conclusion

In this paper, we provide the first comprehensive review of gradient-based multi-objective deep learning, a field of growing importance as models are required to balance multiple, often conflicting, objectives. We have systematically surveyed the spectrum of algorithms, covering methods for finding a single balanced model, a finite set of models, and an entire infinite Pareto set of models.

The review also delves into the theoretical foundations of these methods, summarizing key results in convergence while highlighting the need for a deeper understanding of generalization. The practical significance of MOO is demonstrated across diverse applications, including its emerging role in Large Language Models (LLMs). By unifying these approaches and identifying key challenges, this paper serves as a foundational resource to guide and inspire future advancements in this critical and rapidly evolving domain.

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
