# OpenReview forum: "Gradient-Based Multi-Objective Deep Learning: Algorithms, Theories, Applications, and Beyond"
_TMLR — Under review for TMLR_

### Review · Reviewer_zaW3 · 2026-05-17

**Summary Of Contributions:**

This paper provides a survey of gradient-based methods for multi-objective deep learning. Only supervised learning is surveyed, no RL. Multi-objective is defined as the simultaneous minimization of a set of objective functions, with preferences specified across objectives via a probability simplex.

It provides a taxonomy and surveys:
* methods that find a single, well-balanced solution (Sec. 3)
* methods that generate a finite set of diverse Pareto-optimal solutions (Sec. 4)
* methods that learn a continuous Pareto set of solutions (Sec. 5)

Strength: comprehensive survey, does what it claims to do. Coverage seems reasonably broad and the taxonomy is sensible.

Weakness: it reads as a listing of papers, no comparison, no in-depth discussion of strengths and weaknesses of the methods, especially when the different methods are for solving the same problem (3 listed in their taxonomy). The paper does occasionally mention that some methods provide "better performance" but does not go into detail. Better on which benchmarks? Over which methods? What about methods that are not compared?

Weakness: mentions the survey differs from existing surveys on multi-task learning, but the MOO framing is close enough that the distinction needs to be made explicit.

Overall, as a listing of recent papers on MOO it's fine, and quite comprehensive. But does not provide much value beyond.

**Audience:**

Yes

**Audience Explanation:**

It would be interesting to someone looking for a comprehensive listing recent MOO work on supervised learning using deep learning.

**Claims And Evidence:**

Yes

**Claims Explanation:**

It's a survey, and it does survey and categorize papers into its taxonomy, so the claims are supported. Section 2 provides a precise framework for MOO and some technical background. Sections 3, 4, 5 respectively survey methods that fall within the 3 categories in their taxonomy.

**Requested Changes:**

[major] Need to add all relevant details on survey methodology. What are inclusion criteria for papers? Rejection? Cutoff date? Without this it's hard to tell whether coverage is representative or just what the authors happened to read.

[major] Need a bit more editorialization. At the moment it's a listing of methods that solve the same problem, but there must be weaknesses and strengths for each? When to apply which to what? Some recommendations would be helpful. Comparisons on shared benchmarks? Even a table comparing methods within each taxonomic category on a few axes (compute cost, scaling with number of objectives, preference handling) would be helpful.

[major] Whenever the paper says a method provides "better performance," specify on which benchmarks and against which baselines. Right now these claims are vague and unsupported.

[minor] Clarify distinction with prior surveys on multi-task learning. In what sense is the MOO framework different? Which works covered here would not appear in an MTL survey?

---

> ### Author Response · Authors · 2026-06-20
> **Response to Reviewer zaW3**
>
> Thank you for the constructive review. All manuscript changes are highlighted in blue.
>
> **Comment 1:** "[major] Need to add all relevant details on survey methodology. What are inclusion criteria for papers? Rejection? Cutoff date? Without this it's hard to tell whether coverage is representative or just what the authors happened to read."
>
> **Response:** We added a "Survey Methodology" subsection. It specifies paper sources, representative search keywords, the cutoff date of March 31, 2026, inclusion criteria for supervised gradient-based MOO methods, and exclusion criteria for methods designed exclusively for reinforcement learning, Bayesian optimization, or purely gradient-free/evolutionary search.
>
> **Comment 2:** "[major] Need a bit more editorialization. At the moment it's a listing of methods that solve the same problem, but there must be weaknesses and strengths for each? When to apply which to what? Some recommendations would be helpful. Comparisons on shared benchmarks? Even a table comparing methods within each taxonomic category on a few axes (compute cost, scaling with number of objectives, preference handling) would be helpful."
>
> **Response:** We added more editorialization in the method sections rather than only adding a high-level summary. Specifically, each of the three main method sections now includes a comparison table and an expanded discussion of strengths, weaknesses, and when to use which method family. The single-solution section compares loss balancing, gradient weighting, and gradient manipulation by backward-pass cost and update mechanism, then gives recommendations based on objective count, model size, and severity of gradient conflict. The finite-set section compares preference handling, exact preference alignment, and whether solutions are optimized independently or jointly, then recommends preference-vector methods, preference-free methods, or \(m>n\) formulations for different settings. The infinite-set section compares hypernetworks, preference-conditioned networks, and model-combination methods by parameter overhead, scalability, and expressiveness.
>
> **Comment 3:** "[major] Whenever the paper says a method provides "better performance," specify on which benchmarks and against which baselines. Right now these claims are vague and unsupported."
>
> **Response:** Thank you for pointing this out. We revised the manuscript to make performance-related statements more precise and easier to verify. Comparative claims are now either tied to the cited paper's experimental setting or rewritten as method descriptions. For example, RW is described as obtaining comparable performance with twelve MTL baselines on five image datasets and two multilingual tasks from XTREME, as reported in its original paper. Hyper-Trans is described as reporting improvements over MLP-based hypernetworks in its Pareto-set learning experiments. For LORPMAN, we focus on its parameter-efficient Pareto-manifold structure rather than making a broad performance claim. We also replaced wording such as "better conditioning ability" with a mechanism-level description of how FiLM lets preferences modulate intermediate feature channels.
>
> **Comment 4:** "[minor] Clarify distinction with prior surveys on multi-task learning. In what sense is the MOO framework different? Which works covered here would not appear in an MTL survey?"
>
> **Response:** We expanded the related-work section. The revised text explains that standard MTL surveys usually focus on task relatedness, sharing structures, and learning paradigms for one shared model, while the MOO framing centers on objective vectors, partial orders, preference specification, Pareto stationarity, and finite or continuous Pareto-set approximation. We also added concrete examples that are not naturally covered by standard MTL surveys, including preference-vector finite-set methods, hypervolume or particle-based Pareto-set methods, and continuous Pareto-set learning methods such as PHN, LORPMAN, and Panacea.

---

> > ### Comment · Reviewer_zaW3 · 2026-06-21
> >
> > Thank you for making the revisions and improving the quality of the survey.

---

### Review · Reviewer_Wd2n · 2026-05-26

**Summary Of Contributions:**

The submission provides a comprehensive review of gradient-based multi-objective learning. It spends a majority of its space reviewing a substantial set of gradient-based MOO algorithms, drawing a historical throughline of improvements in each family of approaches. It additionally spends some time highlighting key theoretical questions in the domain before referencing a set of applications, datasets, and software packages.

**Additional Comments:**

I have some minor nits:

1. It might be useful to spell out that the preference vector is a simplex because otherwise the problem would be indeterminate (or similar language).
2. I appreciate that the discussion of things like the Pareto front follows the mathematical definition with an informal description. I think the discussion of the Hypervolume indicator would benefit from the same, possibly with a visual illustration similar to Fig. 3. And I believe that "hypervolume indicator" is the more standard term here rather than hypervolume more generally which sometimes just means volume in higher dimension.
3. It would be useful if Fig. 5 could be pushed down so it does not appear beside the single solution method.
4. How are multiple solutions obtained from MGDA in Fig. 5a considering that it is a single solution method?
5. In the definition in Section 5.2, does $\tilde{g}_{\alpha}$ have to be differentiable for things to work? That seems not generally true (e.g. for non-relaxed Tchebycheff scalarization), unless I'm missing something.
6. In section 7.2, the DARTS parenthetical citation has double parentheses, and the later citations (beginning with Wu et al.) are parenthetical but I think should not be. Perhaps a different bibtex command would be appropriate here?

**Audience:**

Yes

**Audience Explanation:**

The core question with any survey like this is whether it's more than just a list of references, and thus useful to readers in TMLR's audience. From this perspective, I think the submission mostly succeeds when it comes to review of algorithms and approaches (though structure and clarity could be improved as I discuss below). Carrying throughlines from MGDA to CAGrad and later PMTL and the other MGDA variants, for example, provides a useful consistent perspective on a family of methods developed over time. Likewise the taxonomy in Figure 2 is a useful value-add to understanding the field, and the setup in earlier sections that enables a shared discussion of optimization in section 5.2 is thoughtful and well-done.

I have some concerns that the same thoughtfulness is not extended to the later sections in the paper (applications, datasets, packages etc seem somewhat perfunctory relative to the earlier discussion). These sections could either be cut (to focus on the methods where the paper does provide a useful shared perspective), or expanded (to match the structure of the rest of the paper).

If they are expanded, I think a missing component is discussion of what is actually interesting / challenging etc in those areas, i.e. what unifies the work there and what are the active challenges for MOO. For example, I imagine that in dense prediction problems a challenge is output dimensionality whereas in LLMs a major challenge is parameter scale, and perhaps in NAS a core challenge is awkward search spaces whereas in RecSys it is noisy objective functions. This is just speculation, but my point is that for this part of the survey to be useful we need to see more than just a list of application papers and their citations, there needs to be some useful additional perspective. Software packages could likewise be discussed in context of what are practical engineering challenges and how they are solved rather than a simple list one could find by searching the web, and dataset discussion could be folded alongside applications.

With the earlier (methods) sections, I think a useful improvement would be to bring some of the connections between methods into tables or figures and out of prose / text. For example, Table 2 summarizes the convergence results cleanly apples-to-apples in a way complementary to the discussion in prose, but there is nothing like this for the other sections (Fig. 4-6 give some intuition about method differences, but not quite as obvious as "method X is like method Y except with change Z"). One way of doing this may be to take the middle- or leaf-level nodes of Fig 2 and revisit them in each section in additional detail (either breaking out into more levels, or making explicit "the X -> add Y -> becomes Z" type of logic). I'm not sure if all of this will fall neatly into a shared taxonomy, but my overall point is that most readers would be using this paper as a reference rather than reading it end-to-end, so quick-reference signposts would be helpful.

Finally, in any discussion of gradient-based optimization and deep learning, I think it would be very useful to include some unified discussion of computational complexity. Especially considering the early discussion of how gradient-methods scale better than evolutionary ones, it begs the question of which methods scale to very large networks as used in current practice and which do not (e.g. I assume things like bilevel optimization or anything involving full Hessians might be a nonstarter).

**Broader Impact Concerns:**

No concerns.

**Claims And Evidence:**

Yes

**Claims Explanation:**

I'm reviewing this from the perspective of the TMLR guidance which suggests that surveys should draw new connections, highlight trends, or suggest new problems. On this front, the soundness bar is generally easy to clear for a survey paper and I don't see any major issues here. My one quibble is that the discussion of "Distributed training" under Section 9 says "current MOO algorithms are limited to single GPUs or machines" , which seems very surprising to me considering earlier discussion in the paper of algorithms applied to LLMs. I don't know of any non-toy LLM that an be trained on a single GPU, and even single-node training is more focused on research problems (like NanoGPT) than anything practical. I think authors should carefully justify this or moderate their claims here.

**Requested Changes:**

See above -- I strongly encourage authors to consider either expanding or cutting the shorter and more perfunctory section on applications / datasets / software, and consider the other improvements which may help a reader navigate the survey as a compact reference. All would strengthen the paper in my eyes.

One critical item is the claim that "Current MOO algorithms in deep learning are limited to single GPUs or machines" -- if this is indeed true this is a significant limitation on any practically sized network. But I don't think this is true, as most sharding strategies for distributed training would be method-agnostic and hook at the autograd part of the stack. On the other hand, most methods that work on single machines with multiple GPUs are already distributed and would work multi-node. I do believe methods relying on bilevel optimization or second-order autograd would potentially be more problematic in the distributed setting, as things like double backprop / differentiable layers (for bilevel optimization) or distributed hvp (for second-order methods) are often poorly supported or an area of active research -- but this distinction is a subtlety missing in present discussion.

---

> ### Author Response · Authors · 2026-06-20
> **Response to Reviewer Wd2n [1/2]**
>
> Thank you for the careful review. All manuscript changes are highlighted in blue.
>
> **Comment 1:** "One critical item is the claim that "Current MOO algorithms in deep learning are limited to single GPUs or machines" -- if this is indeed true this is a significant limitation on any practically sized network. But I don't think this is true, as most sharding strategies for distributed training would be method-agnostic and hook at the autograd part of the stack. On the other hand, most methods that work on single machines with multiple GPUs are already distributed and would work multi-node. I do believe methods relying on bilevel optimization or second-order autograd would potentially be more problematic in the distributed setting, as things like double backprop / differentiable layers (for bilevel optimization) or distributed hvp (for second-order methods) are often poorly supported or an area of active research -- but this distinction is a subtlety missing in present discussion."
>
> **Response:** This is a fair concern. We revised the distributed-training paragraph to moderate the original statement. The revised text says that many studies report single-GPU or single-machine experiments, but this is not an inherent incompatibility with distributed training. We now discuss the specific scaling issues raised here: communicating per-objective gradients, synchronizing multiple models or preference-conditioned branches, and scaling bilevel or second-order computations such as hypergradients.
>
> **Comment 2:** "I have some concerns that the same thoughtfulness is not extended to the later sections in the paper (applications, datasets, packages etc seem somewhat perfunctory relative to the earlier discussion). These sections could either be cut (to focus on the methods where the paper does provide a useful shared perspective), or expanded (to match the structure of the rest of the paper)."  "See above -- I strongly encourage authors to consider either expanding or cutting the shorter and more perfunctory section on applications / datasets / software, and consider the other improvements which may help a reader navigate the survey as a compact reference. All would strengthen the paper in my eyes."
>
> **Response:** Thank you for the constructive suggestion. We agree that these sections should not read as a loose list of application papers, datasets, and software. We chose to keep them because they play a complementary role to the method-centered sections: they show how the earlier taxonomy appears in common deep learning settings, and they give readers a compact entry point for benchmarks and tools.
>
> Within the limited revision time, we made a focused expansion rather than rewriting these parts as standalone domain surveys. In the applications section, we revised the framing so that each domain is presented in terms of the multi-objective trade-offs it illustrates and its connection to the earlier algorithmic taxonomy. In the resources section, we added selection rationale for the datasets and libraries, emphasizing the objective structures covered by the datasets and the practical roles of LibMTL and LibMOON for single-solution optimization and Pareto-set exploration. This keeps the paper centered on gradient-based MOO methods while making the later sections more useful as compact reference material. We also view the reviewer's suggestion as a valuable direction for future versions of the survey, where these sections could be further expanded with more detailed domain-specific discussions.
>
> **Comment 3:** "With the earlier (methods) sections, I think a useful improvement would be to bring some of the connections between methods into tables or figures and out of prose / text."  "my overall point is that most readers would be using this paper as a reference rather than reading it end-to-end, so quick-reference signposts would be helpful."
>
> **Response:** We agree that quick-reference signposts are important for a survey. The manuscript already provides an overall taxonomy figure, and we now add focused comparison tables inside the three main method sections. The new tables compare representative methods for (i) finding a single solution, (ii) finding a finite solution set, and (iii) learning an infinite Pareto set. They summarize per-iteration cost, preference handling, alignment behavior, coupling among solutions, parameter overhead, scalability, and representative mechanisms, so readers can use the paper more easily as a reference.

---

> ### Author Response · Authors · 2026-06-20
> **Response to Reviewer Wd2n [2/2]**
>
> **Comment 4:** "Finally, in any discussion of gradient-based optimization and deep learning, I think it would be very useful to include some unified discussion of computational complexity."
>
> **Response:** We added a unified computational-complexity discussion in the single-solution methods section and complemented it with the new comparison tables. For finite- and infinite-set methods, the new tables and discussions emphasize the main scaling factors and practical trade-offs: preference-vector methods are easy to parallelize but depend on the chosen trade-offs, preference-free methods jointly optimize multiple candidates, and infinite-set architectures trade expressiveness against parameter overhead and scalability.
>
> **Comment 5:** "It might be useful to spell out that the preference vector is a simplex because otherwise the problem would be indeterminate (or similar language)."
>
> **Response:** We added this explanation in the preference-vector definition in Section 2.1: the simplex constraint makes preferences nonnegative and normalized, so they express relative trade-off importance rather than arbitrary rescaling.
>
> **Comment 6:** "I appreciate that the discussion of things like the Pareto front follows the mathematical definition with an informal description. I think the discussion of the Hypervolume indicator would benefit from the same, possibly with a visual illustration similar to Fig. 3. And I believe that "hypervolume indicator" is the more standard term here rather than hypervolume more generally which sometimes just means volume in higher dimension."
>
> **Response:** We updated the terminology to "hypervolume indicator" and added intuition in Section 2.1. In two-objective minimization, it is the area dominated by the obtained objective vectors and bounded by the reference point; in higher dimensions, it generalizes to dominated volume.
>
> **Comment 7:** "It would be useful if Fig. 5 could be pushed down so it does not appear beside the single solution method."
>
> **Response:** We moved the figure to the beginning of the finite-set section, immediately after the section introduction.
>
> **Comment 8:** "How are multiple solutions obtained from MGDA in Fig. 5a considering that it is a single solution method?"
>
> **Response:** We clarified this in the caption. The multiple MGDA points are obtained from independent MGDA runs with different random model-parameter initializations. MGDA itself remains a single-solution method and does not include an explicit diversity-promoting mechanism.
>
> **Comment 9:** "In the definition in Section 5.2, does $\tilde g_\alpha$  have to be differentiable for things to work? That seems not generally true (e.g. for non-relaxed Tchebycheff scalarization), unless I'm missing something."
>
> **Response:** We clarified the notation. $\tilde g_\alpha$ denotes a preference-dependent scalar training criterion induced by a scalarization or MOO solver. Gradient-based training uses a differentiable criterion or differentiable/smoothed surrogate; nonsmooth criteria such as standard Tchebycheff scalarization can be handled with subgradients or smooth approximations.
>
> **Comment 10:** "In section 7.2, the DARTS parenthetical citation has double parentheses, and the later citations (beginning with Wu et al.) are parenthetical but I think should not be. Perhaps a different bibtex command would be appropriate here?"
>
> **Response:** We corrected the citation formatting by changing the DARTS reference to a standard citation.

---

### Review · Reviewer_y84K · 2026-06-06

**Summary Of Contributions:**

This survey focuses of gradient-based multi-objective optimization for deep learning (MOO).

1. It examines how deep learning frameworks can simultaneously optimize multiple competing objectives.

2. It offers a very well structured review of algorithms, theoretical foundations (including theoretical convergence guarantees), real-world applications, benchmarking studies. Each section is followed by an easy-to-digest discussion section providing a high-level summary. It is also a useful reminder that many learning problems are inherently multi-objective.

3. It discusses key unresolved challenges in the field.

**Audience:**

Yes

**Audience Explanation:**

I think the paper is quite relevant to several communities including people working on general multi-task learning, and people working on alignment and fairness.

Most machine learning practitioners are accustomed to optimizing a single scalar objective. The survey serves as a useful reminder that many learning problems are inherently multi-objective, and that modern methods can do more than find a single compromise solution: they can approximate an entire Pareto front, allowing users to navigate trade-offs after training.

**Claims And Evidence:**

Yes

**Claims Explanation:**

The focus of the paper is on the gradient-based MOO methods for supervised deep learning. For this scope, paper seems to be quite solid and complete.

It introduces a problem and covers methods for:

- finding a single solution,

- finding a finite set of solutions,

- finding an infinite Pareto set,

It additionally surveys convergence and generalization guarantee as well as applications and datasets.

I do have minor editorial suggestions (and a few questions), but otherwise the paper appears to be quite accurate.

**Requested Changes:**

Eq (1) p1: It is not quite clear how one should minimize a vector function. I would argue that Eq (1) is actually informal and the actual formalizations are, e.g., Pareto-optimal solutions or weighted combinations of the objectives. It looks like this “informality” is common in MOO papers. However, a survey paper should strive for conceptual clarity.

P2. The paragraph that starts with “However, gradient-based MOO faces several significant challenges in deep learning” needs citations to back up multiple claims.



Regarding evolutionary algorithms: This is true for classic methods, but not necessarily for LLMs whose performance can depend a lot on prompting. Evolutionary prompt-optimization algorithms such as GEPA and EvoPrompt can be quite efficient.

To be clear, I think it’s fine to focus only on gradient-based approaches.

EvoPrompt: Connecting LLMs with Evolutionary Algorithms Yields Powerful Prompt Optimizers. Guo et al.

GEPA: Reflective Prompt Evolution Can Outperform Reinforcement Learning. A Agrawal et al

End p2-p3 : “However, these works primarily treat the problem from a standard multi-task perspective rather than formally grounding the optimization process within the framework of MOO.” Please, elaborate. This is very unclear. What is “framework of MOO”? How is it different from a standard multi-task framework?

Table 1: are decisions variables the same as model parameters?

Eq (9a) has the same issue as Eq (1): we can’t directly optimize the multi-objective function.

Page 10: Shouldn’t Eq (15) have a constant c?

Page 12: “Sener & Koltun (2018) use feature-level gradients (i.e., gradients w.r.t. the representations from the last shared layer) to replace the parameter-level gradients (i.e., gradients w.r.t. the shared parameters θ). Since the dimension of the representation is much smaller than that of the shared parameters.” Does it mean that parameters of the shared layer are not updated?

Page 14: “Two methods for improving the Tchebycheff scalarization include smooth Tchebycheff scalarization, which replaces the non-smooth max operation with a smooth approximation, and Preference-based MGDA (PMGDA), which only requires the update direction to have a negative inner product with the exact constraint gradient.” This paragraph needs citations.

Page 18 and LORPMAN algorithm. I think it’s worth noting a connection to LORA. There are also additional approaches that combine multiple LORA adapters, e.g., mixture of LORAs.

BTW below you mention SVD-LoRA, but without a citation: “such as large language models, using SVD-LoRA:”

Page 19: " PCGrad (Yu et al., 2020) also offer convergence analyses in similar approaches.” the theoretical guarantee of PGGrad is very limited: it is not a general nonconvex multi-task convergence theorem for arbitrary number of tasks. It is a fairly specific theoretical analysis for the two-task setting (with convex objectives and other limitations).

Table 2: Could you confirm that Table 2 estimates are for the same definitions of Pareto-stationarity?

Page 21: “MoDo achieves convergence guarantees without the bounded function value assumption.” Does it completely remove the assumption or replaces it with some other assumptions?

Page 24: I think a more accurate description for CIFAR 100 is “In multi-task learning benchmarks, CIFAR-100 is often partitioned into 20 tasks according **to its 20 coarse categories**, each task being a 5-class classification problem.”

---

> ### Author Response · Authors · 2026-06-20
> **Response to Reviewer y84K [1/2]**
>
> Thank you for the detailed review. All manuscript changes are highlighted in blue.
>
> **Comment 1:** "Eq (1) p1: It is not quite clear how one should minimize a vector function. I would argue that Eq (1) is actually informal and the actual formalizations are, e.g., Pareto-optimal solutions or weighted combinations of the objectives. It looks like this “informality” is common in MOO papers. However, a survey paper should strive for conceptual clarity."
>
> **Response:** Thank you for highlighting this ambiguity. We clarified the convention at its first occurrence. After Eq. (1), the manuscript now states that minimizing a vector-valued function is MOO shorthand in the standard vector-optimization sense, as discussed in Boyd and Vandenberghe's *Convex Optimization* and classical MOO texts: solutions are compared by Pareto dominance, and algorithms seek Pareto-optimal solutions or scalarized subproblems.
>
> **Comment 2:** "P2. The paragraph that starts with “However, gradient-based MOO faces several significant challenges in deep learning” needs citations to back up multiple claims."
>
> **Response:** We added the requested citations. The introduction-level challenges now cite representative works on preference alignment, Pareto-set approximation cost, and stochastic-gradient noise or biased common descent directions.
>
> **Comment 3:** "Regarding evolutionary algorithms: This is true for classic methods, but not necessarily for LLMs whose performance can depend a lot on prompting. Evolutionary prompt-optimization algorithms such as GEPA and EvoPrompt can be quite efficient. To be clear, I think it’s fine to focus only on gradient-based approaches. EvoPrompt: Connecting LLMs with Evolutionary Algorithms Yields Powerful Prompt Optimizers. Guo et al. GEPA: Reflective Prompt Evolution Can Outperform Reinforcement Learning. A Agrawal et al"
>
> **Response:** This is a helpful distinction. We revised the scope statement to separate direct gradient-free search over neural-network parameters from prompt-space optimization. The manuscript now explicitly notes that EvoPrompt and GEPA operate in discrete prompt space and can be effective there, while our survey remains focused on gradient-based optimization of deep models.
>
> **Comment 4:** "End p2-p3 : “However, these works primarily treat the problem from a standard multi-task perspective rather than formally grounding the optimization process within the framework of MOO.” Please, elaborate. This is very unclear. What is “framework of MOO”? How is it different from a standard multi-task framework?"
>
> **Response:** We agree that the previous wording was too compressed. The revised related-work paragraph now distinguishes MTL and MOO along three axes: the problem object, the output target, and the evaluation criterion. Standard MTL surveys usually organize around tasks, sharing architectures, and learning paradigms for training one shared model or task-head system. MOO instead treats the objective vector and its partial order as central, where no canonical scalar optimum exists before a preference or selection rule is specified. This also changes the output from one model to one compromise, a finite Pareto-set approximation, or a preference-conditioned manifold, and changes evaluation toward Pareto dominance/stationarity, hypervolume, preference alignment, and coverage. We also added concrete examples that are not naturally covered by standard MTL surveys: preference-vector finite-set methods, hypervolume or particle-based Pareto-set methods, and continuous Pareto-set learning methods such as PHN, LORPMAN, and Panacea.
>
> **Comment 5:** "Table 1: are decisions variables the same as model parameters?"
>
> **Response:** Yes. In deep learning settings, the decision variable $\theta$ usually corresponds to model parameters. We clarified this in the notation table.
>
> **Comment 6:** "Eq (9a) has the same issue as Eq (1): we can’t directly optimize the multi-objective function."
>
> **Response:** Yes, Eq. (9a) follows the same convention as Eq. (1). Since the clarification after Eq. (1) explains how vector-valued objectives should be interpreted in the standard MOO sense, we do not repeat the same explanation after each later vector-valued formulation.
>
> **Comment 7:** "Page 10: Shouldn’t Eq (15) have a constant c?"
>
> **Response:** Thank you for catching this. We added the missing constant $c$ and state that $c \geq 0$ controls the conflict-aversion strength.

---

> ### Author Response · Authors · 2026-06-20
> **Response to Reviewer y84K [2/2]**
>
> **Comment 8:** "Page 12: “Sener & Koltun (2018) use feature-level gradients (i.e., gradients w.r.t. the representations from the last shared layer) to replace the parameter-level gradients (i.e., gradients w.r.t. the shared parameters θ). Since the dimension of the representation is much smaller than that of the shared parameters.” Does it mean that parameters of the shared layer are not updated?"
>
> **Response:** We clarified this point to avoid misunderstanding. Feature-level gradients are used only to compute objective weights or update directions. The shared parameters are still updated by backpropagating the resulting scalarized or aggregated loss through the network.
>
> **Comment 9:** "Page 14: “Two methods for improving the Tchebycheff scalarization include smooth Tchebycheff scalarization, which replaces the non-smooth max operation with a smooth approximation, and Preference-based MGDA (PMGDA), which only requires the update direction to have a negative inner product with the exact constraint gradient.” This paragraph needs citations."
>
> **Response:** We added the missing citations for smooth Tchebycheff scalarization and PMGDA.
>
> **Comment 10:** "Page 18 and LORPMAN algorithm. I think it’s worth noting a connection to LORA. There are also additional approaches that combine multiple LORA adapters, e.g., mixture of LORAs. BTW below you mention SVD-LoRA, but without a citation: “such as large language models, using SVD-LoRA:”"
>
> **Response:** Thank you for pointing out this connection. We revised the LORPMAN paragraph to use terminology closer to the cited works. The manuscript now states that LORPMAN is related to LoRA-style adaptation, but differs from LoRAHub and Mixture of LoRA Experts (MoLE). LoRAHub combines already-trained LoRA modules for cross-task generalization without additional model parameters or gradients; MoLE treats trained LoRAs as experts and learns gating weights for flexible composition. LORPMAN instead jointly learns a shared full-rank model and preference-weighted low-rank components as a continuous Pareto-manifold subspace, so preferences index points in a learned subspace rather than routed combinations of pre-trained adapters. We also revised the sentence introducing Panacea so that SVD-LoRA is clearly described as Panacea's preference-conditioned low-rank parameterization, with preferences injected through singular values.
>
> **Comment 11:** "Page 19: " PCGrad (Yu et al., 2020) also offer convergence analyses in similar approaches.” the theoretical guarantee of PGGrad is very limited: it is not a general nonconvex multi-task convergence theorem for arbitrary number of tasks. It is a fairly specific theoretical analysis for the two-task setting (with convex objectives and other limitations)."
>
> **Response:** We revised this statement to match the scope of the original analysis. The manuscript now states that PCGrad provides a limited theoretical analysis under specific two-task and convex-objective assumptions.
>
> **Comment 12:** "Table 2: Could you confirm that Table 2 estimates are for the same definitions of Pareto-stationarity?"
>
> **Response:** Yes. The estimates in the convergence table (Table 5 in the revised manuscript; Table 2 in the original submission) are for the same definition of Pareto-stationarity used in the paper.
>
> **Comment 13:** "Page 21: “MoDo achieves convergence guarantees without the bounded function value assumption.” Does it completely remove the assumption or replaces it with some other assumptions?"
>
> **Response:** MoDo removes the bounded-function-value assumption while still relying on $L$-smoothness and bounded stochastic gradients, as summarized in the table. We revised the corresponding text to make this clear.
>
> **Comment 14:** "Page 24: I think a more accurate description for CIFAR 100 is “In multi-task learning benchmarks, CIFAR-100 is often partitioned into 20 tasks according to its 20 coarse categories, each task being a 5-class classification problem.”"
>
> **Response:** We adopted the suggested wording in the dataset subsection.